# Cardiac electrophysiological remodeling associated with enhanced arrhythmia susceptibility in a canine model of elite exercise

Alexandra Polyák[1†], Leila Topal[1†], Noémi Zombori-Tóth[1], Noémi Tóth[1],
János Prorok[1,2], Zsófia Kohajda[2], Szilvia Déri[1], Vivien Demeter-Haludka[1],
Péter Hegyi[3,4,5], Viktória Venglovecz[1], Gergely Ágoston[6], Zoltán Husti[1],
Péter Gazdag[1], Jozefina Szlovák[1], Tamás Árpádffy-Lovas[1], Muhammad Naveed[1],
Annamária Sarusi[1], Norbert Jost[1,2,7], László Virág[1,7], Norbert Nagy[1,2],
István Baczkó[1,7]*, Attila S Farkas[8]*, András Varró[1,2,7]*

[1]Department of Pharmacology and Pharmacotherapy, University of Szeged, Szeged, Hungary; [2]ELKH-SZTE Research Group for Cardiovascular Pharmacology, Eötvös Loránd Research Network, Szeged, Hungary; [3]Centre for Translational Medicine and Institute of Pancreatic Diseases, Semmelweis University, Budapest, Hungary; [4]Institute for Translational Medicine, Medical School, University of Pécs, Pécs, Hungary; [5]Translational Pancreatology Research Group, Interdisciplinary Centre of Excellence for Research Development and Innovation, University of Szeged, Szeged, Hungary; [6]Institute of Family Medicine, University of Szeged, Szeged, Hungary; [7]Department of Pharmacology and Pharmacotherapy, Interdisciplinary Excellence Centre, University of Szeged, Szeged, Hungary; [8]Department of Internal Medicine, Cardiology ward, University of Szeged, Szeged, Hungary

*For correspondence:
baczko.istvan@med.u-szeged.
hu (IB);
farkas.attila@med.u-szeged.hu
(ASF);
varro.andras@med.u-szeged.
hu (AV)

†These authors contributed equally to this work

Competing interest: The authors declare that no competing interests exist.

**Abstract** The health benefits of regular physical exercise are well known. Even so, there is increasing evidence that the exercise regimes of elite athletes can evoke cardiac arrhythmias including ventricular fibrillation and even sudden cardiac death (SCD). The mechanism of exercise-induced arrhythmia and SCD is poorly understood. Here, we show that chronic training in a canine model (12 sedentary and 12 trained dogs) that mimics the regime of elite athletes induces electrophysiological remodeling (measured by ECG, patch-clamp, and immunocytochemical techniques) resulting in increases of both the trigger and the substrate for ventricular arrhythmias. Thus, 4 months sustained training lengthened ventricular repolarization (QTc: 237.1±3.4 ms vs. 213.6±2.8 ms, n=12; APD90: 472.8±29.6 ms vs. 370.1±32.7 ms, n=29 vs. 25), decreased transient outward potassium current (6.4±0.5 pA/pF vs. 8.8±0.9 pA/pF at 50 mV, n=54 vs. 42), and increased the short-term variability of repolarization (29.5±3.8 ms vs. 17.5±4.0 ms, n=27 vs. 18). Left ventricular fibrosis and HCN4 protein expression were also enhanced. These changes were associated with enhanced ectopic activity (number of escape beats from 0/hr to 29.7±20.3/hr) in vivo and arrhythmia susceptibility (elicited ventricular fibrillation: 3 of 10 sedentary dogs vs. 6 of 10 trained dogs). Our findings provide in vivo, cellular electrophysiological and molecular biological evidence for the enhanced susceptibility to ventricular arrhythmia in an experimental large animal model of endurance training.

## Editor's evaluation

This is a comprehensive study in large animals which is a technical/logistical challenge to accomplish. The work is valuable in that exercise is an important part of the lives of a significant proportion of the world's inhabitants. The authors have presented solid data from canines, which are more translatable to humans, showing that arrhythmogenic remodeling at the cellular and tissue levels are associated with elite exercise training.

## Introduction

The health benefits of regular exercise are well known (*Lawler et al., 2011*; *Quindry and Franklin, 2021*). However, there is increasing evidence that chronic high-level exercise in elite athletes can evoke cardiac arrhythmias including atrial fibrillation (*Aschar-Sobbi et al., 2015*; *Baldesberger et al., 2008*) and even sudden cardiac death (SCD; *Maron and Pelliccia, 2006*). For the better understanding of this complex issue, it may be helpful to be familiar with the U-shaped relationship between exercise intensity and the risk of adverse cardiovascular events (*Merghani et al., 2016*); although both moderate- and vigorous-intensity exercise improve health, exercise at a competitive level may be an additional potential risk factor for arrrhythmogenesis in certain individuals or in situations where repolarization reserve is impaired. It is usually due to frequently silent underlying conditions, such as hypertrophic cardiomyopathy, long QT syndrome, diabetes, electrolyte imbalances, doping, or otherwise harmless presumed medications.

Fortunately, sport-related SCD is rare, although its incidence may often be underestimated. Thus, SCD seems to be 2.8 times more frequent in elite athletes than in age-matched populations *Corrado et al., 2008* who do not engage in sporting activity. It has to be mentioned that SCD incidence was found much higher than that in certain young athletes population such as male college basketball players (*Raukar et al., 2017*). In addition, in only a few cases, the cause of death has been satisfactorily established by autopsy findings; part of the remaining cases of SCD has been attributed to ventricular fibrillation of ischemic origin. However, the latter explanation can be challenged because, very often, SCD in elite athletes does not occur during peak performance when oxygen demand is indeed very high in the myocardium. Instead, SCD occurs during warmup, after exercise, or even at home during rest. Therefore, the cause and mechanism of SCD due to heavy chronic exercise should be also sought elsewhere.

It was recently reported that high levels of exercise in rats or mice induce electrophysiological remodeling resulting in atrial fibrillation (*Aschar-Sobbi et al., 2015*; *Guasch et al., 2013*), sinus bradycardia (*D'Souza et al., 2015*) and atrioventricular (AV) node dysfunction (*Mesirca et al., 2021*). Also, in a recent study in rats after high-intensity chronic exercise, it was found that exercise-trained rats developed eccentric cardiac hypertrophy, together with both left ventricular (decreased S-wave in pulmonary vein flow and increased left ventricular isovolumic relaxation time) and right ventricular (decreased E-wave velocity and prolonged E-wave deceleration time) diastolic dysfunction and with atrial enlargement. Also, collagen deposition in the right ventricle was significantly higher, which were associated with enhanced vulnerability to arrhythmia at the supraventricular level (*Benito et al., 2011*). The mechanism of SCD in human elite athletes is, for obvious reasons, very hard to study, while animal studies which were focused on AF after chronic endurance training were carried out in mice or in rats in which cardiac function, e.g., heart rate and repolarization properties, differs in important ways from human (*Aschar-Sobbi et al., 2015*; *Benito et al., 2011*; *D'Souza et al., 2014*; *Guasch et al., 2013*). With the exception of one incomplete and preliminary study (*Polyák et al., 2018*) and a very recent report on AV dysfunction in race horses (*Mesirca et al., 2021*), there have been no experimental studies of the subject in large animals which would have better translational value (*Varró et al., 2021*). Accordingly, the aim of the present study was to determine the effect of 4 months of sustained exercise on cardiac remodeling and possible arrhythmia susceptibility in a canine model that better reflects human physiology and pathophysiology.

# Results

## Sustained exercise-induced cardiac hypertrophy and fibrosis

All echocardiographic parameters and autopsy outcomes are presented in *Table 1*. The 16 weeks of endurance training led to increased left atrial volume (LAV) in the trained group, suggesting left atrial enlargement. Additionally, thickening of the interventricular septal (IVS) and left ventricular posterior walls (LVPW), greater end-diastolic diameter (LVEDD) in the left ventricle, and increased left ventricular mass and left ventricular mass index (LVM and LVMi), as signs of left ventricular hypertrophy, is also developed in trained animals. End-diastolic left ventricular volume (EDV) was also increased in the trained dogs compared to the sedentary controls. The differences between the groups persisted after normalization of the measured echocardiographic parameters to body weight (IVS/BW; LVPW/BW) or BSA (LAVi; left ventricular end-systolic diameter [LVESD]/BSA; LVEDD/BSA; EDV/BSA). End-systolic volume (ESV), its BSA adjusted value (ESV/BSA), and the ejection fraction (EF) did not differ between the examined groups.

In parallel with the in vivo echocardiographic findings, autopsy outcomes showed cardiac hypertrophy, with an increment in LVMi, the thickening of the IVS (IVS/BW), and LVPW (LVPW and LVPW/BW) in trained canine hearts compared to sedentary hearts. In addition, some degree of enhanced fibrosis was also present in the left ventricle of the trained canine hearts compared to the sedentary hearts (*Figure 1*).

## Effect of sustained training on the heart rate

Sustained training resulted in significant bradycardia (*Figure 2A*) and increase in heart rate variability parameters (rmsSD-RR, root mean square of Successive Differences of RR interval) in the TRN dogs (*Figure 2B*) compared to the sedentary controls and their baseline values at 0th week. Intrinsic beating rate was also examined on spontaneously beating isolated right atrial tissue preparations from SED and TRN dogs to observe changes without the dependence of the autonomic nervous system. The spontaneous rate was significantly slower in the trained canine tissue preparations than in the sedentary subjects (*Figure 2C*), further suggesting that the bradycardia observed in trained dogs is not entirely due to enhancement of the vagal tone.

## ECG changes and increased proarrhythmic response following long-term sustained training

The 16 weeks of sustained training prolonged the RR, PQ, QT, and heart-rate corrected QT (QTc) intervals, moreover lengthened the TpTe interval and widened the QRS complex on the electrocardiogram in conscious, trained dogs (*Table 2*). The prolonged QT interval was also associated with significantly increased QT interval variability ('short-term variability' of the QT intervals; STV-QT), reflecting the elevated dispersion of repolarization measured after completion of the training protocol in the trained animals compared to sedentary controls (*Table 2*).

In conscious, SED animals, there were only a few ventricular beats during the 3×20 min ECG recordings at rest. However, a significantly higher incidence of ventricular extrasystole was observed in the TRN animals (*Figure 3B*), and the majority of which were ventricular escape beats (*Figure 3A and C*). The incidence of premature beats comparing to that of the escape beats was lower in both experimental groups, and it did not differ significantly between the TRN and SED animals (*Figure 3C*). More complex types of arrhythmias were not observed in either group at rest.

During electrical burst stimulation in open-chest anesthetized dogs, ventricular fibrillation was elicited in 6 out of 10 TRN dogs, whereas ventricular fibrillation occurred in only 3 out of 10 of the SED control dogs (*Figure 3D and E*).

## Influence of sustained training on cardiac ventricular action potentials

Cardiac ventricular action potentials were measured in both isolated left ventricular papillary muscle preparations representing subendocardial origin and in enzymatically isolated left ventricular single myocytes representing midmyocardial origin. As *Figure 4B* shows, the cardiac action potential duration measured as 90% repolarization (APD$_{90}$) of left ventricular papillary muscle preparations did not differ significantly between the examined groups; however, in the enzymatically isolated left

**Table 1.** Echocardiographic parameters before and after 16-week-long vigorous training in canine hearts and autopsy outcomes after heart removal.

The effect of 16-week-long vigorous training on echocardiographic cardiac parameters and performance in canine hearts and autopsy investigations after heart removal. Echocardiographic values were measured before (at 0th week, control measurements) and after (at 16th week) the training protocol. IVS, interventricular septum; BW, body weight; LVPW, left ventricular posterior wall thickness; LVESD, left ventricular end-systolic dimension; LVEDD, left ventricular end-diastolic dimension; LVM, left ventricular mass; LVMi, left ventricular mass index; EDV, end-diastolic volume; ESV, end-systolic volume; EF, ejection fraction; LAV, left atrial volume; LAVi, left atrial volume index. Data are expressed as mean ± SEM.

**Echocardiographic parameters before and after long-term vigorous training**

| | Before the training protocol (at 0th week) | | After the training protocol (at 16th week) | |
| --- | --- | --- | --- | --- |
| | 'SED' group | 'TRN' group | 'SED' group | 'TRN' group |
| IVS, mm | 7.1±0.3 | 6.8±0.2 | 7.4±0.2 | 8.13±0.2*† |
| IVS/BW, mm/kg | 0.6±0.03 | 0.5±0.03 | 0.6±0.03 | 0.74±0.03*† |
| LVPW, mm | 7.1±0.2 | 6.95±0.2 | 7.4±0.3 | 7.64±0.3 |
| LVPW/BW, mm/kg | 0.6±0.03 | 0.6±0.03 | 0.6±0.03 | 0.70±0.04*† |
| LVESD, mm | 14.2±0.3 | 17.6±0.7 | 18.7±0.5 | 18.4±0.1 |
| LVESD/BSA, mm/m$^2$ | 25.8±1.1 | 32±1.03 | 34.2±1.5 | 36.5±2.1† |
| LVEDD, mm | 28.7±0.7 | 29±0.96 | 30.4±0.7 | 32.0±0.7† |
| LVEDD/BSA, mm/m$^2$ | 51.7±1.4 | 52.7±1 | 55.3±1.9 | 63.5±1.3*† |
| LVM, g | 46.7±3.4 | 45.1±2.1 | 54.1±3.9 | 63.6±2.8† |
| LVMi, g/m$^2$ | 83.6±5.6 | 81.2±2.8 | 97.7±6.4 | 125.8±4.3*† |
| EDV, ml | 32.3±2 | 32.6±2.5 | 37.9±2.2 | 40.6±1.7† |
| EDV/BSA, ml/m$^2$ | 57.5±2.5 | 58.4±2.9 | 68.4±3.4 | 80.3±2.3*† |
| ESV, ml | 6.1±0.6 | 9.5±1.03 | 11.4±0.8 | 10.2±1.3 |
| ESV/BSA, ml/m$^2$ | 10.9±0.9 | 17±1.5 | 20.7±1.3 | 20.1±2.5 |
| EF, % | 80.4±1.7 | 70.9±2.3 | 69.7±1.3 | 75.1±2.7 |
| LAV, ml | 9±0.9 | 8.9±0.6 | 10.4±0.9 | 11.4±1.4† |
| LAVi, ml/m$^2$ | 16±1.4 | 16±0.8 | 18.7±1.3 | 22.4±2.3† |

**Autopsy findings after the long-term vigorous training**

| | IVS, mm | IVS/BW, mm/kg | LVPW, mm | LVPW/BW, mm/kg | LVM, g | LVMi, g/m$^2$ |
| --- | --- | --- | --- | --- | --- | --- |
| 'SED' group | 3.62±0.6 | 0.028±0.04 | 2.54±0.4 | 0.02±0.03 | 79.3±4.4 | 144.1±4.1 |
| 'TRN' group | 4.25±0.3 | 0.039±0.03* | 3.42±0.3* | 0.031±0.02* | 83.3±4.8 | 167.2±5.7* |

*p<0.05 'TRN' vs. 'SED' group at 16th week by unpaired Student's t-test.

†p<0.05 'TRN group at 16th week' vs. 'TRN group at 0th week' by paired Student's t-test.

The online version of this article includes the following source data for table 1:

**Source data 1.** The effect of chronic endurance training on echocardiographic cardiac parameters and autopsy outcomes.

**A**

Fibrosis score 0
Sedentary dog

Fibrosis score 1
Trained dog

Fibrosis score 2
Trained dog

**B**

**C**

**Figure 1.** Increased level of myocardial fibrosis in the canine left ventricle. (**A**) Representative histological images of left ventricular free-wall connective tissue visualized by Crossmon's trichrome staining taken from SED dogs (fibrosis score 0=negative outcome) and the TRN dogs (fibrosis score 1=mild and fibrosis score 2=moderate level of fibrosis). Black arrows indicate the presence of fibrosis. (**B**) Bar chart showing the incidence of fibrosis, expressed as the percentage of the total number of animals, irrespective of the degree of fibrosis. (**C**) Bar chart estimating the amount of scarring via fibrosis scoring in SED and TRN dogs. The 'n' numbers refer to the number of dogs included. Data are expressed as mean ± SEM. *$p<0.05$ 'TRN' vs. 'SED' group at 16th week by chi-square test.

The online version of this article includes the following source data for figure 1:

**Source data 1.** The effect of chronic endurance training on the incidence of fibrosis.

**Source data 2.** The effect of chronic endurance training on the level of fibrosis.

ventricular myocytes from TRN dogs, the $APD_{90}$ significantly lengthened (***Figure 4A and C***) and STV-APD increased (***Figure 4D and E***) compared to SED animals.

## Possible influence of sustained training on cardiac ventricular transmembrane currents

As ***Figure 5A*** shows, the magnitude of the $I_{to}$ current was significantly smaller in myocytes obtained from the chronically trained dogs compared to the sedentary animals. However, there were no significant differences in the magnitudes of the $I_{NaL}$ (***Figure 5B***), $I_{NCX}$ (***Figure 5C***), L-type $I_{Ca}$ (***Figure 5D***), $I_{K1}$ (***Figure 5E***), $I_{Kr}$ (***Figure 5F***), and $I_{Ks}$ currents (***Figure 5G***).

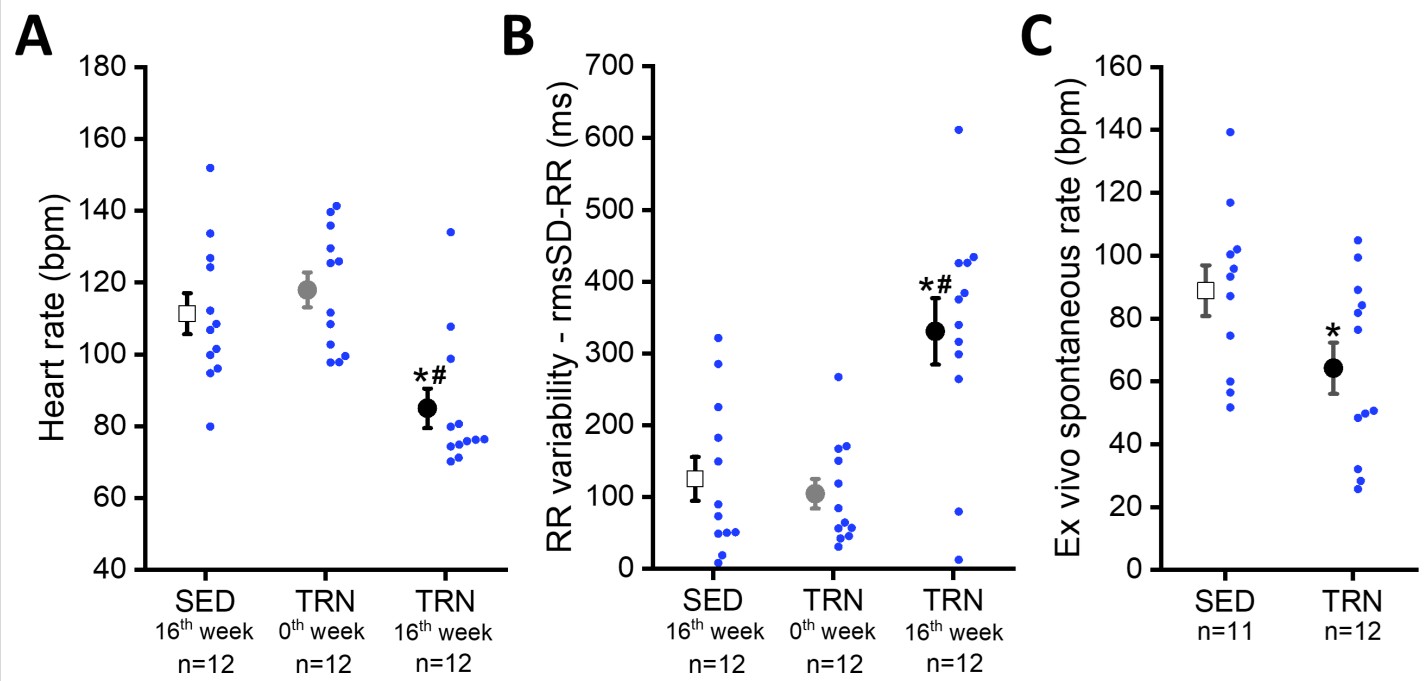

**Figure 2.** Effect of sustained training on heart rate and heart rate variability in conscious dogs and on spontaneous beating rate in isolated right atrial tissue preparations. (**A**) Demonstrates the bradycardia and (**B**) demonstrates the heart rate variability values of conscious dogs in the SED group at the end of the training protocol (at 16th week; n=12 dogs), in the TRN group before the chronic endurance training (at 0th week; n=12), and in the TRN group after the chronic endurance training (at 16th week; n=12 dogs). RmsSD-RR, root mean square of Successive Differences of RR interval. (**C**) The spontaneous beating rate in isolated right atrial tissue preparations obtained from SED (n=11) dogs and TRN (n=12) dogs. The 'n' numbers refer to the number of dogs included. Data are expressed as mean ± SEM. Blue dots represent individual data. *$p<0.05$ 'TRN' vs. 'SED' group at 16th week by unpaired Student's t-test. #$p<0.05$ 'TRN' group at 16th week vs. 'TRN' group at 0th week by paired Student's t-test.

The online version of this article includes the following source data for figure 2:

**Source data 1.** Effect of chronic sustained training on the heart rate in conscious dogs.

**Source data 2.** Effect of chronic sustained training on the RR variability in conscious dogs.

**Source data 3.** Effect of chronic sustained training on the spontaneous beating rate in isolated right atrial tissue preparations.

## The amount of transmembrane Kv4.3 and KChiP2 proteins in trained and sedentary canine hearts

Further studies were performed on the molecular nature of the reduced $I_{to}$ current in the TRN dog heart. Specifically, the expression of the Kv4.3 alpha and KChiP2 beta channel subunits, which are considered to be the most important channel proteins underlying $I_{to}$, was studied by western blotting and immunocytochemistry measurements. As **Figure 6A–F** indicate, no significant differences in Kv4.3 and KChiP2 protein expression were found between the left ventricular samples of the SED and TRN dog hearts, suggesting that decrease in the magnitude of $I_{to}$ current may be due to the changes in other less well-characterized accessory proteins or alternatively to post-translational changes in ion channel proteins.

## Upregulation of left ventricular HCN4 channels in TRN dogs after sustained training

Immunocytochemistry of myocytes obtained from the TRN dog left ventricle showed significantly enhanced HCN4 protein expression compared to the SED dogs (**Figure 7A**). There were no differences in the expression of HCN1 and HCN2 proteins between the two groups (**Figure 7B and C**).

**Table 2.** Electrocardiographic parameters before and after 16-week-long vigorous training in conscious, sedentary, and trained dogs.

Electrocardiographic values were measured before (at 0th week, control measurements) and after (at 16th week) the training protocol. QTc, heart-rate corrected QT interval; STV-QT, 'short-term variability' of the QT intervals. Data are expressed as mean ± SEM.

**Electrocardiographic parameters before and after long-term vigorous training**

|  | Before the training protocol (at 0th week) | | After the training protocol (at 16th week) | |
| --- | --- | --- | --- | --- |
|  | 'SED' group (n=12) | 'TRN' group (n=12) | 'SED' group (n=12) | 'TRN' group (n=12) |
| RR, ms | 588.4±32.1 | 579.3±33.2 | 644.2±58.6 | 841.8±62.8*[†] |
| PQ, ms | 103.2±2.0 | 98.3±2.9 | 102.8±3.2 | 110.7±3.6[†] |
| QRS, ms | 59.6±1.7 | 60.5±2.4 | 56.3±2.6 | 70.8±1.6*[†] |
| QT, ms | 218.7±5.7 | 215.9±2.9 | 223.0±6.4 | 251.3±3.2*[†] |
| QTc, ms | 216±4.7 | 213.6±2.8 | 217.7±4.5 | 237.1±3.4*[†] |
| STV-QT, ms | 2.6±0.2 | 2.5±0.2 | 2.6±0.2 | 3.6±0.4*[†] |
| TpTe, ms | 27.3±2.3 | 27.9±2.5 | 30.9±2.4 | 36.5±1.7[†] |

*p<0.05 'TRN' vs. 'SED' group at 16th week by unpaired Student's t-test.

[†]p<0.05 'TRN group at 16th week' vs. 'TRN group at 0th week' by paired Student's t-test.

The online version of this article includes the following source data for table 2:

**Source data 1.** Effect of chronic endurance training on ECG parameters in conscious dogs at rest.

## Discussion

This study is an extension of our previous animal athlete's heart model studies of long-term endurance exercise in various species (*Kui et al., 2021*; *Polyák et al., 2018*; *Topal et al., 2022*), providing further insight into the electrophysiological properties of the athlete's heart and the associated increased ventricular arrhythmic risk.

The main findings of the study are: (1) endurance training increased heart rate variability, which indicates an increased parasympathetic tone, and decreased resting heart rate in both whole animal and in vitro experiments, suggesting that there may also be factors beyond the increased vagal tone that influence the development of training-related bradycardia. (2) Morphological adaptations characteristic of the athlete's heart, such as left ventricular hypertrophy, increased atrial and ventricular volumes also developed. (3) Sustained training increased the expression of HCN4 channels in the ventricular myocardium, which, in parallel with increased levels of fibrosis, may contribute to the increased arrhythmia sensitivity observed in this model. (4) Following chronic endurance training, repolarization prolongation was a consistent observation, manifesting as moderate, but significant QTc prolongation in conscious dogs and APD prolongation accompanied with reduced $I_{to}$ current density in cellular measurements. These findings were associated with increased variability of cardiac repolarization. (5) Also, sustained exercise training appeared to increase the risk of fibrillation in ventricles subjected to electrical stimulation. Our results imply an increased sensitivity to arrhythmias in trained canine hearts. Considering also the limitations, our canine model with the applied exercise protocol may be a useful experimental model to further investigate cardiovascular effects of long-term endurance exercise.

### Animal models of the human athlete's heart

There is an increased incidence of unexpected death among elite athletes (*Corrado et al., 2008*), the exact cause of which is poorly defined and, as such, is an open issue (*Raukar et al., 2017*). Our working hypothesis proposed the question whether these tragic events are related to changes in ventricular repolarization and also to fibrosis that may develop following prolonged high-intensity exercise in competitive athletes (*Domenech-Ximenos et al., 2020*; *Małek and Bucciarelli-Ducci, 2020b*). However, investigating the relevant mechanisms in humans is difficult for several reasons, and therefore, appropriate animal models with human relevance are needed. The majority of animal

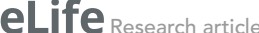

**Figure 3.** Incidence of ventricular extra beats and ventricular fibrillation in sedentary and trained dogs after the 16-week-long vigorous training. (**A**) Representative ECG recordings of escape beats and increased R–R interval variability at rest in a conscious TRN dog at 16th week. (**B**) Bar chart plotting the increased incidence of different ventricular extra beats (premature and escape) at rest in the TRN group vs. SED group at 16th week. (**C**) Reports the average of premature and ventricular escape beats across animals at rest in conscious SED and TRN dogs at 16th week. (**D**) Representative burst pacing ECG image in anesthetized SED and TRN dogs at 16th week. (**E**) Bar chart plotting the increased incidence of ventricular fibrillation (VF) in the TRN group vs. SED group following burst arrhythmia provocation. The 'n' numbers refer to the number of dogs included. Data are expressed as mean ± SEM. Blue dots represent individual data. *p<0.05 'TRN' vs. 'SED' group at 16th week by Mann-Whitney U test.

The online version of this article includes the following source data for figure 3:

**Source data 1.** The incidence of ventricular extra beats in conscious trained and sedentary dogs at 16th week.

**Source data 2.** The number of ventricular extra beats in conscious trained and sedentary dogs at 16th week.

**Source data 3.** The incidence of ventricular fibrillation in anesthetized trained and sedentary dogs at 16th week.

experiments in this field have so far been conducted in small animal models. In rodents, for example, electrophysiological remodeling, cardiac fibrosis, development of bradycardia, and increased risk of atrial fibrillation have been reported after exercise training (*Aschar-Sobbi et al., 2015*; *Benito et al., 2011*; *D'Souza et al., 2014*; *Gazdag et al., 2020*; *Guasch et al., 2013*; *Oh et al., 2020*). While it is undeniable that these experiments have revealed several important aspects about the athlete's heart, these species differ from humans in important aspects of cardiac electrophysiology; for example, the expression pattern of ion channels differs greatly between species and tissue types which strongly limits or even eliminates their translational value to humans. While some chronic exercise models have been reported using rabbits (*Polyák et al., 2018*; *Yuan et al., 2018*), no such studies have

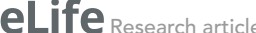

**Figure 4.** Effects of long-term sustained training on cardiac action potential duration and short-term variability in left ventricular preparations of sedentary and trained dogs. *Left panel* of (**A**) representative action potential curves recorded from isolated left ventricular myocytes of SED and TRN dogs, showing the prolonged action potential duration in TRN dog. *Central (SED)* and *right (TRN)* panels of (**A**) 30–30 representative action potential curves indicating the increased variability of the action potential duration in TRN animals recorded from ventricular myocytes, respectively. (**B**) Graph showing the action potential duration measured at 90% repolarization (APD$_{90}$) in SED and TRN dogs recorded from papillary muscle (multicellular tissue preparations; n=48 preparations/12 dogs in SED and n=51 preparations/12 dogs in TRN groups). (**C**) Graph showing the prolonged APD$_{90}$ of enzymatically isolated left ventricular single myocytes in TRN dogs (n=25 cells/7 dogs in SED and n=29 cells/5 dogs in TRN groups). (**D**) Representative Poincare plot of increased APD$_{90}$ variability. (**E**) Effect of long-term sustained training on APD$_{90}$ variability of a single left ventricular myocyte (n=18 cells/7 dogs in SED and n=27 cells/5 dogs in TRN groups). The 'n' numbers refer to the number of preparations or cells followed by the number of dogs from which preparations or cells were obtained. Data are expressed as mean ± SEM. Blue dots represent individual data. *p<0.05 'TRN' vs. 'SED' group at 16th week by unpaired Student's t-test.

The online version of this article includes the following source data for figure 4:

**Source data 1.** Effect of sustained training on the cardiac action potential duration (APD$_{90}$) obtained from multicellular tissue preparations.

**Source data 2.** Effect of sustained training on the cardiac action potential duration (APD$_{90}$) obtained from left ventricular single myocytes.

**Source data 3.** Effect of sustained training on the short-term variability of the action potential duration obtained from left ventricular single-cell myocytes.



**Figure 5.** Effects of chronic sustained training on various transmembrane ionic currents in canine left ventricular myocytes. (**A**) Effect of sustained training on transient outward potassium current ($I_{to}$); representative current recordings (*left*) and current-voltage relationships (*right*) in SED and TRN subjects. (**B and C**) Graphs indicating that sustained training has no effect on either the late $Na^+$ current ($I_{NaL}$) or the $Na^+/Ca^{2+}$ exchange current ($I_{NCX}$). (**D–G**) The current-voltage relationships of the L-type $Ca^{2+}$ current ($I_{CaL}$), the inward rectifier $K^+$ current ($I_{K1}$), and the rapid ($I_{Kr}$) and the slow ($I_{Ks}$) delayed rectifier $K^+$ currents are similar for SED and TRN subject. Insets show the voltage protocols. The 'n' numbers refer to the number of cells followed by the number of dogs from which cells were obtained. Data are expressed as mean ± SEM. Blue dots represent individual data. *$p<0.05$ 'TRN' vs. 'SED' group at 16th week by unpaired Student's t-test and Mann-Whitney U test.

The online version of this article includes the following source data for figure 5:

**Source data 1.** Effects of sustained training on the current-voltage relationship of the transient outward potassium current.

**Source data 2.** Effects of sustained training on the late $Na^+$ current.

**Source data 3.** Effects of sustained training on the $Na^+/Ca^{2+}$ exchange current.

**Source data 4.** Effects of sustained training on the current-voltage relationship of the L-type $Ca^{2+}$ current.

**Source data 5.** Effects of sustained training on the current-voltage relationship of the inward rectifier $K^+$ current.

**Source data 6.** Effects of sustained training on the current-voltage relationship of the rapid delayed rectifier $K^+$ current.

**Source data 7.** Effects of sustained training on the current-voltage relationship of the slow delayed rectifier $K^+$ current.

been performed in dogs or other large animals with high relevance to human cardiac electrophysiology. Available data in canine models indicate that both acute (*Babai et al., 2002*) and adequate levels of chronic exercise (*Bonilla et al., 2012*; *Holycross et al., 2007*) are beneficial in preventing acute ischemia-induced ventricular fibrillation through delayed preconditioning (*Babai et al., 2002*) or restoration of the distorted parasympathetic–sympathetic balance observed after myocardial

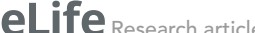

**Figure 6.** Protein expression and relative density of Kv4.3 and KChIP2 subunits determined by western blotting and immunocytochemistry in left ventricular samples from SED and TRN dogs. (**A and B**) Relative protein expression of Kv4.3 and KChIP2 subunits determined by western blotting in left ventricular samples of SED (n=12) and TRN (n=12) dogs, respectively. (**C**) Representative image of Kv4.3 and KChIP2 bands and their corresponding loading controls (GAPDH). (**D and E**) Relative densities of Kv4.3 and KChIP2 protein immunolabeling obtained from SED (n=30 cells/6 dogs) and TRN (n=30 cells/6 dogs) cardiomyocytes. (**F**) Representative immunofluorescence images of canine cardiomyocytes with Kv4.3 and KChIP2 immunolabeling. The 'n' numbers refer to the number of dogs (**A and B**) or the number of cells followed by the number of dogs from which the cells were obtained (**D and E**). Data are expressed as mean ± SEM. Blue dots represent individual data.

The online version of this article includes the following source data for figure 6:

**Source data 1.** Relative protein expression of Kv4.3 subunit determined by western blotting in sedentary and trained dogs.

**Source data 2.** Relative protein expression of KChiP2 subunit determined by western blotting in sedentary and trained dogs.

**Source data 3.** Relative density of Kv4.3 subunit determined by immunocytochemistry in sedentary and trained dogs.

**Source data 4.** Relative density of KChiP2 subunit determined by immunocytochemistry in sedentary and trained dogs.

**Source data 5.** Original unedited membranes of western blots with the relevant bands clearly labeled.

**Source data 6.** Original files of the full raw unedited membranes of western blots.

infarction (***Billman, 2009***; ***Holycross et al., 2007***). Although these data confirm the effects of exercise on the cardiovascular and autonomic nervous systems, the responses to training at above-optimal levels by elite athletes are still unclear. Limited data are available on cardiac electrophysiological remodeling after sustained exercise in sled dogs, and these data are mostly derived from in vivo ECG recordings of long-term racing sled dogs and their controls. They revealed a widened QRS complex



**Figure 7.** Effects of long-term vigorous training on HCN1, HCN2, and HCN4 protein expression determined by immunocytochemistry in enzymatically isolated left ventricular myocytes of SED and TRN dogs. (**A**) Relative density of HCN4 protein immunolabeling obtained from enzymatically isolated left ventricular myocytes of SED (n=30 cells/6 dogs) and TRN (n=30 cells/6 dogs) groups. (**B and C**) Chronic training had no effect on the relative density of HCN1 and HCN2 proteins immunolabeling obtained from left ventricular myocytes of SED (n=30 cells/6 dogs) and TRN (n=30 cells/6 dogs) groups. (**A, B, and C**) *on the right* are representative original immunofluorescence images. The 'n' numbers refer to the number of cells followed by the number of dogs from which the cells were obtained. Data are expressed as mean ± SEM. Blue dots represent individual data. *$p<0.05$ 'TRN' vs. 'SED' group at 16th week by unpaired Student's t-test.

The online version of this article includes the following source data for figure 7:

**Source data 1.** Effect of chronic training on HCN4 protein expression determined by immunocytochemistry in sedentary and trained dogs.

**Source data 2.** Effect of chronic training on HCN1 protein expression determined by immunocytochemistry in sedentary and trained dogs.

**Source data 3.** Effect of chronic training on HCN2 protein expression determined by immunocytochemistry in sedentary and trained dogs.

and a prolonged QT interval (*Constable et al., 1994*; *Constable et al., 2000*); however, no increased arrhythmogenesis or cellular electrophysiological alternations have been investigated in association with these changes.

To the best of our knowledge, this is the first study to provide comprehensive experimental data on arrhythmic electrophysiological remodeling and the associated increased arrhythmic risk at the ventricular level in a canine model of sustained chronic endurance training. Based on our findings, we propose a novel hypothesis based on enhanced arrhythmia substrate and trigger on the mechanism of SCD associated with vigorous endurance exercise, according to which severe ventricular arrhythmias are associated with alterations in cardiac repolarization that may be observed after chronic high-intensity endurance exercise.

## Cardiac morphological adaptation to long-term exercise

Since the frequency and the specific nature of each sporting activity have a fundamental influence on the exercise-induced physiological cardiac response and adaptation (*Morganroth et al., 1975*), it was essential to validate the cardiac morphological changes induced by the exercise modality applied. In this model, the structural changes were left ventricular chamber enlargement with increased

wall thickness occurring in the trained group, as confirmed by both echocardiography and autopsy measurements. However, similar to human data, the EF remained unchanged (*Pelliccia et al., 2010*; *Tahir et al., 2019*). Our structural and hemodynamic outcomes typically correlate with the previously described exercise-induced cardiac remodeling in elite endurance athletes (*Grazioli et al., 2015*; *Pluim et al., 2000*). Although the limitation of the model is that it does not correlate perfectly with any human physical activity, the change in left ventricular architecture suggests that it is close to that of 'isotonic exercise' activities (e.g., long-distance running, cycling, and swimming), as both ventricular size and LVM increased, similar to chronic volume overload (*Kim and Baggish, 2016*).

Our findings of trained dogs correspond to highly trained athletes, who also develop moderately enlarged LAV and LAV index (LAVi), as a potential physiological adaptation to exercise conditioning (*D'Andrea et al., 2010*). Although our work has focused primarily on changes in the left ventricle, it is important to highlight that morphological changes involving the atria may also be the origin of arrhythmias such as atrial fibrillation that is commonly observed in athletes (*Sorokin et al., 2011*).

## Bradycardia and the potential underlying mechanisms

In the present study, prolonged PQ intervals and significant bradycardia were found in ECG recordings following chronic exercise, the latter also persisted under in vitro conditions. Although bradycardia is a general and well-established finding in elite athletes and in animal models of endurance exercise (*Billman et al., 2015*; *Gourine and Ackland, 2019*), its exact mechanism is still a matter of debate (*Billman, 2017a*; *Billman, 2017b*; *Boyett et al., 2017*; *Gourine and Ackland, 2019*). The most common interpretation attributes bradycardia to increased vagal tone both in athletes (*Gourine and Ackland, 2019*) and in animal exercise studies (*Billman et al., 2015*; *Flannery et al., 2017*) as well. However, this theory has generated some disagreement (*Coote and White, 2015*; *D'Souza et al., 2014*; *D'Souza et al., 2015*), and a more satisfactory approach remains to be explored. A very recent publication by *Mesirca et al., 2021* revealed that the slowing of AV conduction persisted after vegetative blockade in race horses and in mice after swimming-induced exercise. This study further argues for the electrical remodeling of the sinoatrial node (SAN), more specifically the reduction in hyperpolarization-activated 'funny' current ($I_f$) density and the remodeling of the underlying HCN4 ionic channel, as previously published in SAN preparations of mice (*D'Souza et al., 2014*).

The present data partially support the observations of *D'Souza et al., 2014* and *Mesirca et al., 2021*, as ionic mechanisms underlying SAN cell automaticity including the $I_f$ and the SAN hyperpolarization-activated cyclic nucleotide-gated channel (HCN) isoforms have not been investigated in this work. Despite the limited data, however, our model seemed to support the findings of the previously mentioned works, as it revealed a significant degree of sinus bradycardia in isolated right atrial preparations from trained hearts after the termination of the autonomic system. On the other hand, the increased beat-to-beat variability of cycle length and first-degree AV block argue for an important contribution of enhanced vagal tone as well, as these values are considered parameters of parasympathetic activity (*Aubert et al., 2003*) and are especially common among athletes with high aerobic resistance.

It should also be emphasized that the pacemaker function of the SAN is complex and cannot be satisfactorily explained by the $I_f$ current alone. Since activation of $I_f$ occurs largely at voltages more negative than the maximal diastolic potential of SAN cells, the special importance of $I_f$ as the main pacemaker current of the SAN has even been questioned (*Morad and Zhang, 2017*; *Noma et al., 1983*). In addition, other mechanisms based on calcium handling (*Maltsev and Lakatta, 2010*; *Verkerk et al., 2013*; *Vinogradova et al., 2005*) or on the contribution of other ion channels have been proposed (*Hu et al., 2021*; *Kohajda et al., 2019*; *Ono et al., 2003*) to explain the cardiac pacemaker effect of SAN. In summary, our present data suggest that both parasympathetic activity and intrinsic SAN changes are parallel responses to vigorous endurance exercise, irrespective of the still unexplored mechanisms.

## Possible mechanisms of arrhythmias in the canine athlete's heart model

Regardless of the nature of the bradycardia, a slower heart rate would itself result in a longer APD and enhanced dispersion of cardiac repolarization. This may be reflected as an increase in the arrhythmic substrate factor in the classical concept of the 'arrhythmic triangle,' which postulates that arrhythmia appears under a certain combination of substrates, arrhythmia triggers, and arrhythmia-promoting

modulators. In addition, bradycardia resulting in longer diastolic intervals would increase the chance of spontaneous diastolic depolarization reaching the firing threshold, which may act as a potential arrhythmia trigger (*Varró and Baczkó, 2010*).

It should also be emphasized that in the present experiment, significant bradycardia was associated with an increased number of escape beats. Well-trained athletes often have slow heart rates, with occasional sinus pauses and, frequently, multiple escape beats. Escape arrhythmia is considered to be a compensatory mechanism caused by increased vagal tone and/or disturbance in the SAN or other parts of the cardiac conduction system. It can also be interpreted as a form of ectopic pacemaker activity that is unveiled by lack of other pacemakers to stimulate the ventricles. In athletes, however, it is generally considered to be a benign ECG pattern that disappears during exercise as the vagal tone decreases. The morphology of the widened QRS complex associated with the P wave on the ECG also suggests an electrical conduction abnormality (*Crescenzi et al., 2020*). These findings again highlight the fact that vigorous exercise induces significant changes in the cardiac conduction system that are not yet completely understood.

Evidence now suggests that HCN channels are highly expressed in the left ventricle of hypertrophic heart and heart failure, contributing to increased arrhythmogenic activity (*Cerbai et al., 1997*; *Cerbai et al., 2001*). Interestingly, in contrary to the bradycardia observed in vivo and in vitro, a concomitant increase in HCN4 protein expression was also observed in the left ventricular myocytes after chronic exercise in this model. This controversy might be explained by tissue-specific regulation of HCN4 channels in the heart. In parallel with the increased expression of HCN4 protein and other potential arrhythmia promoters discussed previously, the incidence of ventricular electrical stimulation-induced ventricular fibrillation also increased in trained animals.

Another consistent finding in this study was the significantly higher level of fibrosis in the left ventricular muscle of trained dogs. Our histopathological results share similarities with several previously reported human and animal studies after chronic endurance exercise (*Kui et al., 2021*; *Małek and Bucciarelli-Ducci, 2020a*; *Topal et al., 2022*). Since our working hypothesis focused on possible repolarization abnormalities, the molecular mechanisms of enhanced fibrosis were not investigated. However, earlier works have explored this issue in some depth in mice and rats (*Aschar-Sobbi et al., 2015*; *Benito et al., 2011*; *Oh et al., 2020*). Despite the fact that this type of structural abnormality is usually silent by different non-invasive methods, e.g., electrocardiography and echocardiography, it may greatly impact the onset and the modulation of re-entry ventricular arrhythmias, as a potentially dangerous myocardial arrhythmia substrate.

Although the exercise-induced compensatory adaptive mechanisms of the athlete's heart have been considered as a phenomenon completely distinct from pathological conditions, the electrical remodeling observed in our model indicates that such intense exercise may not always be beneficial to cardiovascular system and that increased arrhythmia susceptibility may also develop after vigorous exercise.

## Ventricular repolarization in the canine athlete's heart model

The prolongation of cardiac repolarization was a consistent observation in our experiments and manifested as a prolonged QTc interval in conscious dogs in vivo and also a prolonged APD at the cellular level in isolated left ventricular myocytes in the trained group. This finding was associated with increased short-term variability – STV of QT interval on the ECG and also STV of APD in cellular measurements – suggesting increased spatial and temporal dispersion of repolarization. It is notable that variability changes were detected in both in vivo and in vitro, suggesting that this variability parameter may be a valuable and easily measurable biomarker even for further human studies, as a sign of repolarization abnormalities on the electrocardiogram (*Lengyel et al., 2011*), which is potentially present at the cellular level. Although prolongation of the QTc interval, bradycardia (*Bjørnstad et al., 1991*), and increased spatial and temporal dispersion of repolarization have been reported in athletes, reflected in the prolongation of the Tpeak-Tend interval on ECG and STV-QT (*Lengyel et al., 2011*), their putative association with increased incidence of mortality is unclear. Notably, similar repolarization changes have been reported in the failing (*Kääb et al., 1998*; *Kääb et al., 1996*) or in the hypertrophied heart (*Coppini et al., 2013*).

The possible underlying cellular mechanism of repolarization prolongation in the chronically trained dogs in our model was a reduction in the magnitude of $I_{to}$ current in the midmyocardial myocytes. Its

important role in phase 1 repolarization of the action potential in the left ventricular midmyocardial myocytes is well established. A previous study by our research group also found that inhibition of $I_{to}$ current in the canine subepicardial muscle significantly prolonged the APD (*Virág et al., 2011*). Similar results, i.e., APD prolongation with reduced $I_{to}$ current, were the most consistent findings published in failing canine (*Kääb et al., 1996*) or human (*Kääb et al., 1998*) hearts. In agreement with the transmembrane current data, significant APD lengthening was observed only in myocytes harvested from the left ventricular midmyocardial region, where strong $I_{to}$ is expected (*Antzelevitch, 2001*; *Zicha et al., 2003*) but not in subendocardial left ventricular papillary muscle preparations, where a relatively weak magnitude of $I_{to}$ current was reported (*Zicha et al., 2003*; *Zicha et al., 2004*). Interestingly, under the currently used experimental patch-clamp measuring conditions, the magnitudes of different transmembrane ionic currents, which have a key role in the onset and maintenance of repolarization and the plateau phase of action potential ($I_{Kr}$, $I_{Ks}$, $I_{K1}$, $I_{CaL}$, $I_{NaL}$, and $I_{NCX}$), in native canine left ventricular myocytes showed no significant differences between the examined groups. However, it cannot be excluded that - under different conditions - possible intracellular signaling pathways may influence their function.

The repolarization changes, in combination with the mild fibrosis in the left ventricle, would result in enhanced arrhythmia substrate, providing slightly wider vulnerable window for extrasystoles to provoke severe ventricular arrhythmias.

In the present study, the lower $I_{to}$ current density in trained dogs did not seem to be the result of reduced expression of Kv4.3 alpha or KChIP2 beta accessory channel proteins, as neither western blotting nor immunohistochemistry revealed any difference in protein expression between sedentary and trained heart. Similar results, i.e., a decrease in $I_{to}$ current density without changes in the expression of Kv4.2 and KChIP2 proteins, have been reported earlier in chronically exercised rats (*Stones et al., 2009*). Although Kv4.3 and KChIP2 are considered to be the most important proteins determining $I_{to}$ in the canine heart (*Akar et al., 2005*), it should be emphasized that the expression of other $I_{to}$ accessory proteins not investigated in the present study, such as Kvbeta1, Kvbeta2 (*Patel and Campbell, 2005*; *Pérez-García et al., 1999*), $I_{Na}$ beta1 (*Deschênes and Tomaselli, 2002*), DPP6 (*Nadal et al., 2003*; *Radicke et al., 2005*), and DPP10 (*Jerng et al., 2004*), may also influence the function of the $I_{to}$ channel (*Patel and Campbell, 2005*). In addition, $I_{to}$ may be modulated by the activation of protein kinase A (PKA), protein kinase C (PKC), or both (*Fedida et al., 1993*; *Parker and Fedida, 2001*) and may also change after exercise. Since we have not investigated this issue in the present study, some effect of PKA and PKC modulation also cannot be ruled out.

## Limitations

Arrhythmias associated with the athlete's heart, including ventricular fibrillation, have a very low incidence, and the underlying cardiac electrophysiological alterations are relatively modest. In addition, due to the limited capacity of our research laboratory, in this study, we had to focus on the main transmembrane ionic currents, whereas other cellular mechanisms like intracellular calcium handling or calcium-dependent chloride, potassium, and $I_f$-pacemaker currents were not investigated. The same was true for the molecular biological mechanism of enhanced myocardial fibrosis. Also, in order to mimic these changes in experimental animal models, chronic experiments over several months with a relatively large number of animals were required. This is not particularly difficult with small animals like mice or rats, but it is more complicated with large animals like the dog. Such canine experiments are not only costly but, more importantly, require substantial facilities and trained personnel. Although the translational value of canine experiments is high (*Nánási et al., 2021*) and better than that of mice or rat, important electrophysiological differences regarding the repolarization reserve between the canine and human heart are still known (*Jost et al., 2013b*). Taking into consideration that the maximum number of animals used in each training session is usually less than in rodent studies, the advantage of the canine experiments itself is also their limitation. Another possible limitation is that elite athletes train for several years to reach peak performance, but in this study, we only used 4 months of intense training, and translating the exercise duration between species is also difficult. Also, athlete's cardiac arrhythmias and SCD cannot be attributed only to compensated cardiac hypertrophy, but rather to a combination of several co-existing factors such as hypertrophic cardiomyopathy, different kinds of drugs, doping agents, and hypokalemia, which have not been investigated in the present study. As further limitations we think at the time of our study, no dedicated software

was available at the Department of Pathology for the fully quantitative measurement of the fibrotic area per total area on digitized slides; therefore, a widely accepted semi-quantitative method for the evaluation of fibrosis was used by a pathologist blinded to the treatment of the animals. Also, the complexity of the present study and the capacity of our laboratory precluded the assessment of right ventricular myocardium and the experimental setup and sequence of in vivo and in vitro experiments led us to use open chest procedures and epicardial stimulation for the induction of ventricular arrhythmias instead of using a transvenous catheter approach to reduce the time passed until the isolation of cardiac tissue and cells for in vitro experiments.

## Conclusion

The present study suggests that vigorous endurance training by chronic treadmill exercise results in prolongation of cardiac repolarization and increased repolarization instability associated with mild ventricular fibrosis in the canine model of the human athlete's heart. This does not necessarily indicate that at a competitive level, endurance exercise is harmful since the evidences regarding the beneficial effect of exercise are overwhelming. However, in certain individuals or in situations where the repolarization reserve is impaired due to hidden diseases, such as hypertrophic cardiomyopathy, long QT-syndromes, diabetes or electrolyte imbalances, doping substances, or drugs, the observed changes in repolarization and mild fibrosis induced by endurance training in our study may present additional potential risk factors to be considered in the prevention of possible adverse events in competitive sport.

# Materials and methods

## General methods

Animal maintenance and research were conducted in accordance with the National Institutes of Health Guide for the Care and Use of Laboratory Animals. All procedures involving animals were approved by the Ethical Committee for the Protection of Animals in Research of the University of Szeged, Szeged, Hungary (approval numbers: I-74-15-2017 and I-74-24-2017) and by the Department of Animal Health and Food Control of the Ministry of Agriculture and Rural Development (authority approval numbers: XIII/3330/2017 and XIII/3331/2017) and conformed to the rules and principles of the 2010/63/EU Directive. The animals were purchased from an experimental animal breeder, Ásotthalom, Hungary (breeder's authority approval number: XXXV/2018) certified by the Department of Animal Health and Food Control of the Ministry of Agriculture and Rural Development, Hungary.

## Experimental set-up and dog training protocol

At the beginning of the training, all animals were at least 12-month-old, and none of the animals were older than 18 months. Before starting the experiments, the dogs were conditioned to the training protocol, where they were familiarized with the research personnel and were given a few minutes of continuous walk on the treadmill to minimize the distress during training. Those animals (two dogs), which did not voluntarily adhere to the exercise protocol, were excluded from further experiments. After the 3-week conditioning period, beagle dogs of either sex, weighing 9–15 kg, were randomized into sedentary (SED, n=12) or trained (TRN, n=12) groups. The TRN animals underwent a 16-week-long training period, while the SED group did not receive any training. Running sessions were performed on a special canine treadmill system (Dogrunner K9 Racer Treadmill, Dendermonde, Belgium) with controllable gradient and speed intensity. During the a 16-week-long training period, TRN animals were trained for 5 days a week for 2×90 min per day at speeds of 12–18 km/hr (gradually increasing protocol) and with 2×50 min interval running per day at fixed speeds of 4 and 22 km/hr. Regular resting periods were applied to maintain proper hydration; however, the total daily training interruption did not exceed 1.5 hr. The training intensity was maintained with the use of 5–14% inclination. The training protocol was tested in preliminary experiments and set at a maximum level that could be performed without distress. After each training session, the dogs received portions of their preferred food as a reward.

As the capacity of our laboratories and canine treadmills inherently limited the experimental group sizes, no a priori sample size estimation from a power calculation was done. We collected as many data as possible, given the limitations of funding for data collection from large animal chronic experiments

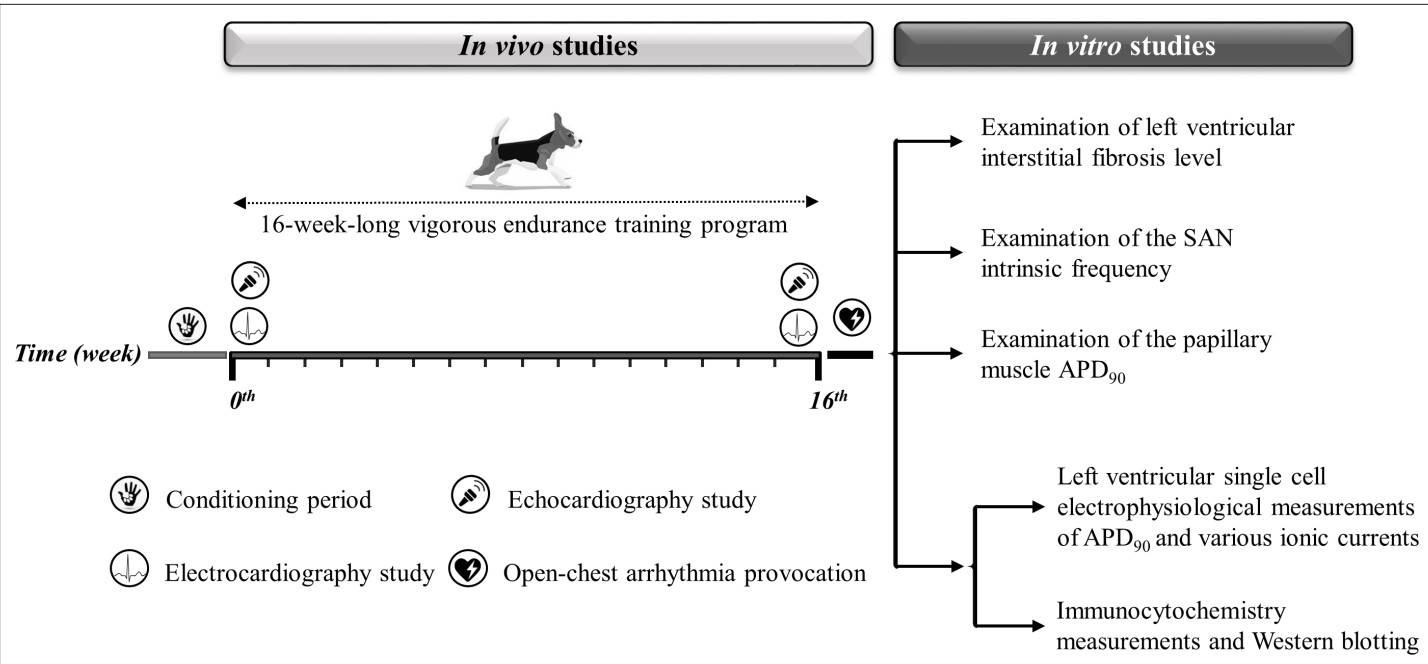

**Figure 8.** Experimental timeline. The experimental timeline of the conditioning and training period and the in vivo and in vitro experiments performed. The symbols illustrate the in vivo studies during the 16-week-long vigorous training. On the right, the in vitro experiments are listed. SAN, sinoatrial node; $APD_{90}$, action potential duration measured at 90% repolarization.

and to minimize the number of animals involved. *Figure 8* and *Table 3* illustrate a detailed experimental timeline.

As far as possible, different stages of experimental processes were carried out blind; the investigators were not aware of the groups when performing the experiments and analyses. However, due to the nature of the experimental settings, in the case of ECG measurements, where recordings were made before ('self-control') and after the 16-week-long training in the same animal, the blinding could not be undertaken.

## Echocardiography

At 0 and 16 weeks of the training protocol, two-dimensional M-mode and Doppler echocardiographic examinations were performed in accordance with the criteria of the American Society of Echocardiography, using 11.5 MHz transducer (GE 10S-RS, GE Healthcare, Chicago, IL, USA), connected to an echocardiographic imaging unit (Vivid S5, GE Healthcare, Chicago, IL, USA). All parameters were analyzed by an investigator in a randomized and blinded manner. The mean values of three measurements were calculated and used for statistical evaluation.

LAV was measured from standard apical four-chamber views at end-systole, and it was corrected for BSA (LAVi). LVESD and LVEDD were measured by means of M-mode echocardiographic images from long-axis and short-axis views between the endocardial borders. Wall thickness parameters (LVPW and IVS) were obtained from parasternal short-axis view and long-axis view. The LVM was calculated using the following formula: LVM (g)=$0.0008 \times \{(1.04 \times [LVEDD + LVPW + IVS]^3 - LVEDD3)+0.6\}$. The LVMi was then calculated by dividing the LVM by the BSA.

IVS and LVPW parameters were also normalized to BW; LVESD, LVEDD parameters, ESV, and EDV to BSA. The BW of the animals was measured immediately before echocardiography. The applied BSA calculation method for dogs has been described earlier by *Corrado et al., 2008*.

The applied ECHO device using a built-in software package calculated the EF according to the Teicholz formula: $EDV = (7 / [2.4+LVEDD]) \times LVEDD^3$; $ESV = (7 / [2.4+LVESD]) \times LVESD^3$; Left ventricular $EF = (EDV − ESV) / EDV$.

**Table 3.** The detailed training protocol.

| Training type Training duration Training speed Training distance | | Long-distance running 2×90 min 12–18 km/hr *gradually increasing speed* (km/day) | Interval running 2×50 min 4 and 22 km/hr *fixed speed* (km/day) | Incline (%) a.m./p.m. | Daily distance (km/day) |
|---|---|---|---|---|---|
| | 1 | 9 | - | - | 9 |
| | 2 | 15 | - | - | 15 |
| | 3 | 22 | - | 4/- | 22 |
| | 4 | 31 | - | 6/- | 31 |
| | 5 | 31 | 7 | 8/4 | 38 |
| | 6 | 33 | 7 | 8/4 | 40 |
| | 7 | 37 | 7 | 10/6 | 45 |
| | 8 | 37 | 7 | 12/8 | 45 |
| | 9 | 39 | 11 | 12/10 | 50 |
| | 10 | 39 | 20 | 12/10 | 59 |
| | 11 | 43 | 20 | 12/10 | 63 |
| | 12 | 46 | 24 | 12/10 | 69 |
| | 13 | 46 | 24 | 12/10 | 69 |
| | 14 | 46 | 25 | 14/12 | 70 |
| | 15 | 46 | 25 | 14/14 | 70 |
| Weeks of training | 16 | 58 | 25 | 14/14 | 82 |

## Electrocardiography

In conscious dogs, ECG recordings were measured using precordial leads at 0 and at 16 weeks. The methods have been described by *Polyák et al., 2018*. In brief, ECGs were recorded simultaneously with National Instruments data acquisition hardware (PC card, National Instruments, Austin, TX, USA) and SPEL Advanced Haemosys software (version 3.26, Experimetria Ltd. and Logirex Software Laboratory, Budapest, Hungary).

RR, PQ, QRS, QT, and $T_{peak}$-$T_{end}$ (TpTe) intervals were measured by manual positioning on-screen markers of 40 consecutive sinus beats at the 10th minute after initiation of the recording, then mean values were calculated. Heart rate was calculated from the RR interval.

As QT interval is influenced by the heart rate, baseline data for ventricular heart rates and QT intervals were used to determine the relationship between the RR interval and the QT interval in sinus rhythm according to *Kui et al., 2016*. Simple linear regression revealed a positive correlation between QT and RR intervals (QT = 0.045 × RR +187). The equations were rearranged to allow the calculation of the rate-corrected QT interval at an RR interval of 528 ms (i.e., a ventricular rate of 118 beats per minute) using the formula: $QTc_x = QT_x - 0.045 \times (RR_{(x-1)} - 528)$. With these equations, plotting QTc against the corresponding RR interval produces a regression line with a slope of zero, indicating that these corrections diminish the influence of heart rate.

Beat-to-beat variability and instability parameters of the RR and QT intervals, such as the rmsSD-RR and the STV-QT, were derived from 40 consecutive sinus beats as described previously (*Polyák et al., 2018*).

The ECG recordings were replayed offline, and the incidence of arrhythmias was calculated from three 20 min ECG recordings taken on three consecutive days (3×20 min) at the 16th week. Ventricular tachyarrhythmia definitions of Lambeth Conventions I (*Walker et al., 1988*) were applied together with all other (non-tachyarrhythmia) ventricular arrhythmia definitions of Lambeth Conventions II (*Curtis et al., 2013*). The total number of arrhythmic beats was calculated as the sum of all ventricular arrhythmic beats in any kind of arrhythmia.

## Open-chest arrhythmia provocation

Following 0.5 µg/kg intravenous sufentanyl (Sufentanil Torrex 5 µg/ml; Chiesi Pharmaceuticals GmbH, Vienna, Austria) premedication and 150 mg/kg intravenous pentobarbital (Release 300 mg/ml; WDT, Garbsen, Germany) anesthesia induction, left thoracotomy was performed on all animals (SED, n=10; TRN, n=10); dogs were endotracheally intubated and mechanically ventilated (UGO Basile S.R.L. respirator; Biological Research Apparatus VA Italy). Physiological parameters (non-invasive blood pressure, oxygen saturation, and electrocardiography) were continuously monitored during surgery and experiments (InnoCare-VET Patient Care Monitor; Innomed Medical Inc, Budapest, Hungary). The ECG was recorded using precordial leads and was digitized and stored as described above.

Under pentobarbital anesthesia, a pacemaker electrode (Biotronik Solia S 60; Biotronik Hungary Ltd., Hungary) was positioned epicardially into the left ventricular apex, and the electrode was connected to a pacemaker (Effecta D; Biotronik Hungary Ltd., Hungary). Pacemakers were programmed in VVI mode using Biotronik IC: 4808A-Renamic programmer to prevent the potentially bradycardic effect of general anesthesia leading to hemodynamic instability. However, bradycardia did not occur in any of the animals, so the pacemakers were not activated. Ventricular threshold was measured before arrhythmia induction in all animals. Ventricular pacing was set to three times the measured threshold in unipolar electrode configuration.

Ventricular arrhythmia inducibility and incidence were tested in both groups using the following stimulation protocol: ventricular burst pacing was applied consecutively for 1, 3, 6, and 9 s, using Effecta D pacemaker in unipolar electrode configuration, at a frequency of 800/min, with a threefold threshold epicardially delivered ventricular stimulation into the apex of the left ventricle. The incidence of arrhythmias induced by the detailed protocols in control and trained animals was compared during the experiments.

## Morphometry and histology

At the end of open-chest arrhythmia provocation, the animals were given an intravenous injection of 400 IU/kg sodium heparin and a sedative (xylazine 1 mg/kg, intravenously) and immediately euthanized with pentobarbital sodium (150 mg/kg, intravenously). After the corneal reflex of each dog had disappeared, the hearts were excised. The atria were removed from the hearts, and ventricles were weighed separately. LVMi was calculated by dividing the measured LVM by the BSA. Ventricular (LVPW) and septal (IVS) wall thicknesses were also measured using a digital caliper and were normalized to BW.

Samples were taken from the ventricular free-wall for histology. Paraffin sections were stained with Crossmon's trichrome staining to identify collagen deposition. Semi-quantitative analysis was performed by an independent pathologist who scored the degree of interstitial fibrosis as follows: 0=negative; 1=mild; 2=moderate.

## Conventional microelectrode techniques

Action potentials were recorded in left ventricular papillary muscle preparations obtained from the hearts of the trained and sedentary dogs using the conventional microelectrode techniques previously described in detail (*Jost et al., 2013b*). Briefly, the preparations were mounted in a tissue chamber of 50 ml volume individually. The experiments were performed using a modified Locke's solution containing (in mM): NaCl 128.3, KCl 4, $CaCl_2$ 1.8, $MgCl_2$ 0.42, $NaHCO_3$ 21.4, and glucose 10. The pH of this solution was set between 7.35 and 7.4 when gassed with 95% $O_2$ and 5% $CO_2$ at 37°C. Each preparation was stimulated through a pair of platinum electrodes in contact with the preparation at a constant basic cycle length of 1000 ms. Transmembrane potentials were recorded after 60 min equilibrium time after mounting using conventional glass microelectrodes, filled with 3 M KCl.

The measurements where the resting membrane potential of the recorded action potential was more positive than –70 mV and/or the action potential amplitude was less than 90 mV were excluded from the analyses.

## Patch-clamp measurements

Ventricular myocytes were enzymatically dissociated as described in detail previously (*Jost et al., 2013a*). A single droplet of cell suspension was placed in a transparent recording chamber mounted on the stage of an inverted microscope (Olympus IX51, Olympus, Tokyo, Japan), and individual

myocytes were allowed to settle and adhere to the chamber bottom for at least 5–10 min before superfusion was initiated and maintained by gravity. Only rod-shaped cells with clear striations were used. HEPES-buffered Tyrode's solution (composition in mM: NaCl 144, NaH$_2$PO$_4$ 0.4, KCl 4.0, CaCl$_2$ 1.8, MgSO$_4$ 0.53, glucose 5.5, and HEPES 5.0, at pH of 7.4) served as the normal superfusate.

Micropipettes were fabricated from borosilicate glass capillaries (Science Products GmbH, Hofheim, Germany), using a P-97 Flaming/Brown micropipette puller (Sutter Co, Novato, CA, USA), and had a resistance of 1.5–2.5 MOhm when filled with pipette solution. The membrane currents were recorded with Axopatch-200B amplifiers (Molecular Devices, Sunnyvale, CA, USA) by means of the whole-cell configuration of the patch-clamp technique. The membrane currents were digitized with 250 kHz analog-to-digital converters (Digidata 1440 A, Molecular Devices, Sunnyvale, CA, USA) under software control (pClamp 10, Molecular Devices, Sunnyvale, CA, USA). All patch-clamp experiments were carried out at 37°C.

### Measurement of L-type calcium current

The L-type calcium current (I$_{CaL}$) was recorded in HEPES-buffered Tyrode's solution supplemented with 3 mM 4-aminopyridine. A special solution was used to fill the micropipettes (composition in mM: CsCl 125, TEACl 20, MgATP 5, EGTA 10, and HEPES 10, pH was adjusted to 7.2 by CsOH).

### Measurement of potassium currents

The inward rectifier (I$_{K1}$), the transient outward (I$_{to}$), the rapid (I$_{Kr}$), and the slow (I$_{Ks}$) delayed rectifier potassium currents were recorded in HEPES-buffered Tyrode's solution. The composition of the pipette solution (mM) was: KOH 110, KCl 40, K$_2$ATP 5, MgCl$_2$ 5, EGTA 5, and HEPES 10 (pH was adjusted to 7.2 by aspartic acid). 1 μM nisoldipine was added to the bath solution to block I$_{CaL}$. When I$_{Kr}$ was recorded, I$_{Ks}$ was inhibited by using the selective I$_{Ks}$ blocker HMR-1556 (0.5 μM). For I$_{Ks}$ measurements, I$_{Kr}$ was blocked by 0.1 μM dofetilide and the bath solution contained 0.1 μM forskolin.

### Measurement of late sodium current

The late sodium current (I$_{NaL}$) was activated by depolarizing voltage pulses of 2 s at –20 mV from holding potential of –120 mV with pulsing cycle lengths of 5 s. After incubation with the drug for 5–7 min, the external solution was replaced with a solution containing 20 μM tetrodotoxin(TTX). TTX at this concentration completely blocked the I$_{NaL}$. The external solution was HEPES-buffered Tyrode's solution supplemented with 1 μM nisoldipine, 0.5 μM HMR-1556, and 0.1 μM dofetilide in order to block I$_{CaL}$, I$_{Ks}$, and I$_{Kr}$ currents. The composition of the pipette solution (in mM) was: CsCl 125, TEACl 20, MgATP 5, EGTA 10, and HEPES 10, pH was adjusted to 7.2 by CsOH.

### Measurement of NCX current

For the measurement of the Na$^+$/Ca$^{2+}$ exchanger current (I$_{NCX}$), the method of *Hobai et al., 1997* was applied. Accordingly, the NCX current is defined as a Ni$^{2+}$-sensitive current and measured in a special K$^+$-free solution (composition in mM: NaCl 135, CsCl 10, CaCl$_2$ 1, MgCl$_2$ 1, BaCl$_2$ 0.2, NaH$_2$PO$_4$ 0.33, TEACl 10, HEPES 10, glucose 10 and ouabain 20 μM, nisoldipine 1 μM, and lidocaine 50 μM, at pH 7.4) as described earlier in detail [*Jost et al., 2013a*]. The pipette solution used for recording I$_{NCX}$ contained (in mM) CsOH 140, aspartic acid 75, TEACl 20, MgATP 5, HEPES 10, NaCl 20, EGTA 20, and CaCl$_2$ 10, pH was adjusted to 7.2 by CsOH.

### Measurements of single-cell action potentials

The perforated patch-clamp technique was used to measure the action potentials of isolated left ventricular myocytes from both trained and sedentary animals. The membrane potential was recorded in current clamp configuration. The myocytes were paced with a rapid rectangular pulse (from 0 to 180 mV, 5 ms) at a frequency of 1 Hz to elicit the action potential. A normal Tyrode solution was used as the extracellular solution containing (in mM): 144 NaCl, 0.4 NaH$_2$PO$_4$, 4 KCl, 0.53 MgSO$_4$, 1.8 CaCl$_2$, 5.5 glucose, and 5 HEPES, titrated to pH = 7.4. The patch pipette solution contained (in mM): 120 K-gluconate, 2.5 NaCl, 2.5 MgATP, 2.5 Na$_2$ATP, 5 HEPES, 20 KCl, and titrated to pH 7.2 with KOH. 50 μM β-escin was added to the pipette solution to achieve the membrane patch perforation. Membrane voltage was obtained by using an Axoclamp 1-D amplifier (Molecular Devices, Sunnyvale,



CA, USA) connected to a Digidata 1440 A (Molecular Devices, Sunnyvale, CA, USA) analog-digital converter. The membrane voltage was recorded by Clampex 10.0 (Molecular Devices, Sunnyvale, CA, USA). At least 60 beats were recorded, and the action potential duration was measured at 90% repolarization ($APD_{90}$). The short-term APD variability (STV-APD) was calculated by analyzing 30 consecutive action potentials.

## Western blot analysis of KChIP2 and Kv4.3 proteins

Membrane fractions were isolated from myocardial samples of TRN (n=12) and SED (n=12) dogs taken from left ventricular free-wall using the method described previously (*Papp et al., 2007*). Protein concentrations were determined by the Lowry method, and 20 µg of each sample was then separated on 8% polyacrylamide gels and transferred to polyvinylidene difluoride western blotting membrane. The membrane was blocked with 2.5% non-fat milk for 1 hr at room temperature and immunolabeled overnight at 4°C with anti-KChIP2 (Alomone, #APC-142, RRID:AB_2756744) and anti-Kv4.3 (Alomone, #APC-017, RRID:AB_2040178) primary antibodies diluted 1:1000. This was followed by incubation for 1 hr with Goat anti-Rabbit IgG-HRP (SouthernBiotech, 4030–05, RRID:AB_2687483) secondary antibody at a dilution of 1:8000. Band densities were detected with ECL Prime Western Blotting Detection Reagent (GE Healthcare) and a ChemiDoc Imaging System (Bio-Rad). Equal loading was verified by glyceraldehyde 3-phosphate dehydrogenase (GAPDH) labeling (ThermoFisher, PA1-988, RRID: AB_2107310). The pixel intensity of each band was measured using ImageJ software. Three parallel western blots were performed for statistical analysis.

## Immunocytochemistry of KChIP2, Kv4.3, HCN1, HCN2, and HCN4 proteins

Cardiomyocytes were isolated from left ventricular tissue of TRN (n=6) and SED (n=6) dogs then fixed on glass coverslips with acetone (*Nagy et al., 2009*). Before immunolabeling, samples were rehydrated with calcium-free PBS and blocked for 1 hr with PBST (PBS with 0.01% Tween) containing 2.5% bovine serum albumin at room temperature. After the incubation period, cells were labeled overnight at 4°C with anti-KChIP2 (Alomone, #APC-142, RRID:AB_2756744), anti-Kv4.3 (Alomone, #APC-017, RRID:AB_2040178), anti-HCN1 (Alomone, #APC-056, RRID:AB_2039900), anti-HCN2 (Alomone, #APC-030, RRID:AB_2313726), and anti-HCN4 (Alomone, #APC-052, RRID:AB_2039906) primary antibodies diluted 1:50. The next day, cells were incubated with goat anti-rabbit IgG Alexa Fluor 488 (ThermoFisher, A-11034, RRID:AB_2576217) secondary antibody (dilution: 1:500, ThermoFisher). Fluorescent images were captured by an LSM 880 (Zeiss) laser scanning confocal microscope. Images were quantitatively analyzed by the ImageJ software. Control samples were incubated only with secondary antibodies.

## Statistics

IBM SPSS Statistics V25, Microsoft Excel (Microsoft Office Professional Plus 2016), and Origin software (2021b, OriginLab) packages were used for statistical analysis. Continuous data were expressed as mean ± SEM. Each figure indicates the number of observations made ('n'), representing the biological replicates of the experiment. The 'n' refers to the number of dogs, except for action potential, patch clamp, western blotting, and immunocytochemistry measurements, where it refers to the number of preparations/cells followed by the number of dogs from which preparations/cells were obtained.

After assessing the normality of our data using Kolmogorov–Smirnov test, paired and unpaired Student's t-test were applied to estimate whether there was a statistically significant difference between the means of the self-control or independent group arrangements, respectively. When data did not follow normal distribution, Mann–Whitney U test was applied instead of Student's t-test. When data could be described by a discrete variable, $\chi^2$ test was applied. Data were considered statistically significant when p≤0.05.

## Acknowledgements

This work was supported by the National Research Development and Innovation Office (NKFIH K 135464 to AV, NKFIH FK-142949 and FK-129117 to NN, NKFIH K 128851 to IB, SNN-134497 to VV and GINOP-2.3.2.-15-2016-00047 and TKP2021-EGA-32), the Ministry of Human Capacities Hungary (20391 3/2018/FEKUSTRAT and EFOP-3.6.2-16-2017-00006), the UNKP-20–5-SZTE-165, the Eötvös Loránd Research Network, and by the Albert Szent-Györgyi Medical School institutional grant (SZTE ÁOK-KKA 2021 to LV). The authors also wish to thank Dr. Tamás Zombori, MD, PhD, and the Department of Pathology for their valuable help to perform the histological measurements.

## Additional information

### Funding

| Funder | Grant reference number | Author |
| --- | --- | --- |
| National Research, Development and Innovation Office | NKFIH K 135464 | András Varró |
| National Research, Development and Innovation Office | FK-142949 and FK-129117 | Norbert Nagy |
| National Research, Development and Innovation Office | NKFIH K 128851 | István Baczkó |
| National Research, Development and Innovation Office | SNN-134497 | Viktória Venglovecz |
| National Research, Development and Innovation Office | GINOP-2.3.2.-15-2016-00047 | Alexandra Polyák |
| National Research, Development and Innovation Office | TKP2021-EGA-32 | Norbert Jost |
| Ministry of Human Capacities Hungary | 20391 3/2018/FEKUSTRAT | László Virág István Baczkó András Varró |
| Ministry of Human Capacities Hungary | EFOP-3.6.2-16-2017-00006 | János Prorok |
| Eötvös Loránd Research Network and Albert Szent-Györgyi Medical School institutional grant | SZTE ÁOK-KKA 2021 | László Virág |

The funders had no role in study design, data collection and interpretation, or the decision to submit the work for publication.

### Author contributions

Alexandra Polyák, Conceptualization, Data curation, Formal analysis, Investigation, Writing – review and editing; Leila Topal, Noémi Tóth, János Prorok, Zsófia Kohajda, Szilvia Déri, Vivien Demeter-Haludka, Zoltán Husti, Péter Gazdag, Jozefina Szlovák, Tamás Árpádffy-Lovas, Muhammad Naveed, Data curation, Formal analysis, Investigation; Noémi Zombori-Tóth, Formal analysis, Funding acquisition, Investigation; Péter Hegyi, Conceptualization, Writing – review and editing; Viktória Venglovecz, Funding acquisition, Investigation; Gergely Ágoston, Annamária Sarusi, Investigation; Norbert Jost, László Virág, Data curation, Formal analysis, Supervision, Investigation, Writing – review and editing; Norbert Nagy, Data curation, Formal analysis, Supervision, Funding acquisition, Investigation, Writing – review and editing; István Baczkó, András Varró, Conceptualization, Supervision, Funding acquisition, Investigation, Writing – original draft, Writing – review and editing; Attila S Farkas, Conceptualization, Investigation, Writing – original draft, Writing – review and editing

## Author ORCIDs
Péter Hegyi http://orcid.org/0000-0003-0399-7259
László Virág http://orcid.org/0000-0002-0592-2608
Norbert Nagy http://orcid.org/0000-0002-4557-8442
István Baczkó http://orcid.org/0000-0002-9588-0797
András Varró http://orcid.org/0000-0003-0745-3603

## Ethics

Animal maintenance and research were conducted in accordance with the National Institutes of Health Guide for the Care and Use of Laboratory Animals. All procedures using animals were approved by the Ethical Committee for the Protection of Animals in Research of the University of Szeged, Szeged, Hungary (approval numbers: I-74-15-2017 and I-74-24-2017) and by the Department of Animal Health and Food Control of the Ministry of Agriculture and Rural Development (authority approval numbers XIII/3330/2017 and XIII/3331/2017) and conformed to the rules and principles of the 2010/63/EU Directive.

## Decision letter and Author response

Decision letter https://doi.org/10.7554/eLife.80710.sa1
Author response https://doi.org/10.7554/eLife.80710.sa2

# Additional files

## Supplementary files
• MDAR checklist

## Data availability

All data generated or analysed during this study are included in the manuscript and supporting file; Source Data files have been provided for Figures 1 - 7 and Table 1 and 2.

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
