## [Editor Report]

This is a comprehensive study in large animals which is a technical/logistical challenge to accomplish. The work is valuable in that exercise is an important part of the lives of a significant proportion of the world's inhabitants. The authors have presented solid data from canines, which are more translatable to humans, showing that arrhythmogenic remodeling at the cellular and tissue levels are associated with elite exercise training.

---

## [Decision Letter]

**Decision letter after peer review:**

Thank you for submitting your article "Cardiac electrophysiological remodeling associated with enhanced arrhythmia susceptibility in a canine model of elite exercise" for consideration by *eLife*. Your article has been reviewed by 4 peer reviewers, one of whom is a member of our Board of Reviewing Editors, and the evaluation has been overseen by Balram Bhargava as the Senior Editor. The following individuals involved in review of your submission have agreed to reveal their identity: Varun Malik (Reviewer #2); Devendra K Agrawal (Reviewer #3); Wei-Hsin Chung (Reviewer #4).

*Reviewer #1 (Recommendations for the authors):*

– I am concerned about the message that elite exercise is associated with ventricular arrhythmogenesis and enhanced risk for fibrillation.

– Elite athletes often train for years to reach peak performance. I am wondering what the translational value of 4 months of training in dogs is relative to elite human athletes.

– The enhanced expression of HCN4 in light of reduced heart rate in trained animals is counterintuitive. How do the authors explain this?

– Did the authors collect serum or plasma samples to examine neurohormonal markers? This would aid the autonomic hypothesis postulated in the study which are currently unsupported.

*Reviewer #2 (Recommendations for the authors):*

Thank you for the opportunity to review this manuscript. In general, I feel that this original contribution is one that makes a great start, and the authors are to be congratulated for the study. Much of the recommendations could be revised and further details are as below. There are some methodological issues detailed – I think these should be listed in the limitations. Given that there is very limited data here- I do not feel that these limitations should preclude publication.

I do feel that the scientific data comprising the evidence for ventricular arrhythmogenesis in exercise training would benefit a review -and it may be worthwhile (once revisions have taken place to a satisfactory level) to consider an editorial commentary (which I would be happy to assist with).

1. I would suggest paying attention to the abstract and its format -I cannot see any information to guide authors on the *eLife* submission guide; but typically, it would be best formatted into a clear background, methods, results, and conclusion section.

2. I agree that smaller animal models less helpful – than in canines, however, given this study also has limitations in extrapolation to humans- I would suggest the removal of the statement: Page 2 lines 39-41 (abstract). The limitation of using animal models has been well-discussed in a very good limitations section.

3. Present the values in the abstract in order of mention (lengthened depol = higher value then lower) etc throughout.

4. I feel that the whole manuscript requires careful editing to ensure that each section is explained clearly, and in order, to avoid any confusing sections. Another reader who is less familiar with the experiments would greatly help. I do not believe that this is a major issue- the paper is generally written reasonably, and the subheadings work well.

a. For instance- methods. The section on animals being euthanised can appear last after the in-vivo methods described first. The morphology should be second-last.

b. The discussion also needs careful revision to avoid repetition and to make sure that the discussion appropriately places the work in the current scientific context.

c. The conclusion also requires shortening to contain a crux summary statement (or two).

d. The limitations section is well written.

e. The conclusion needs to be shorter – much of this should be in the discussion. I do like the authors statement on the clearly beneficial effects of exercise -and these apply only to a subset. This should, however, be in the discussion. The conclusion should really be 2 sentences at most.

f. The discussion should ideally begin with a summary of the principal findings and then move to the discussion points.

5. Introduction: the evidence behind AF and VA is well-understood. It may be better to introduce the "U-shaped" curve of the benefits of exercise with lower levels clearly bad and higher levels -associated with AF.

a. Then as you have done, proceed with the population and mechanism data linking exercise training with VA/SCD.

6. Introduction line 82- please summarise the cardiac dysfunction associated with high intensity exercise

7. Is there a consensus as to the definition of high-intensity exercise?

8. What was the conditioning protocol? This should be detailed in the methods.

9. Why were echo parameters such as EF, atrial and ventricular volumes not assessed? Both left and right ventricular volumes would be helpful.

10. Why was fibrosis not quantitatively measured?

a. This is a methodological limitation.

11. Open chest procedures- the protocol needs to be detailed further.

a. How was access specifically obtained?

b. What was used to produce burst pacing and where in the ventricle was the catheter? Or electrode placed? Was pacing performed at a single site or two sites on the ventricle? Why was the timing 1-3 to 6-9 seconds? Were there between animal differences? Did this produce differences between the groups in terms of pacing protocol?

i. A transvenous catheter would have been better for VF induction.

ii. A more refined protocol for VT stimulation with different sites, burst and programmed extrastimuli ((PES) for set durations, at varying CL) and PES at varying coupling intervals down to VERP would have been much better.

12. Statistical analysis – was the data checked for normal distribution? Normality of the data should be assessed, and the appropriate comparison chosen for statistical analysis. Please clarify.

13. Discussion line 560- I am finding it extremely difficult to draw corollaries between the work of Volders (assessing hypertrophy and fibrosis) in atrioventricular block vs exercise-related fibrosis. This animal model is a model of Tdp (acquired) but I do not feel that this work is an extension of, or related to the work by Volders et al. The discussion will require some revision.

14. What are the possible remodelling mechanisms for the electrophysiologic changes and VA vulnerability – simply fibrosis? Or related to bradycardia (whatever the mechanism)? Or is there any other possible mechanism?

a. Autonomic remodelling and other potentials should be discussed.

15. Given the risk of right ventricular dilatation and dysfunction in athletes (a genotype elusive variant of "ARVC" has been seen in athletes)- RV volume, function and mass would be useful.

a. Do animals with VF susceptibility/PVC have RV features? See work by Heidbuchel, La gerche etc.

*Reviewer #3 (Recommendations for the authors):*

Well done job! No comments

*Reviewer #4 (Recommendations for the authors):*

1. Did the dog undergo a gradually increased training load or constant strength of training? Since there are several kinds of athletes with different training strengths. The author should be specified what kind of athlete should be the subject of this animal model.

2. The ECG presented in this manuscript is referred to VENTRICULAR EXTRASYSTOLE beats, but it should be considered as escape beat after the pause period. It would be problematic if the calculation is based on the escape beats rather than the so-called premature ventricular complex. Is there any references supporting this ventricular beat to be called as extrasystole beats? How does the escape beats increase ventricular arrhythmia inducibility? It is also unclear for the reviewer to understand whether the escape beat is measure at week 12 or 16?

3. The ECG was measure after 10 mins after recording. What are the dogs doing at that time. Since the activity will influence the HR activity and ECG.

4. Are these all dogs having the same ventricular extrasystole burden? It would be possible to show the readers the difference between dogs with VF induction and dogs without VF induction.

5. Since the baseline heart rate is different, is it fair to use the pacing rate to compare the VF inducibility? Or should consider use different pacing rate base on the baseline heart rate? please comment.

6. It is unclear for the reviewer to understand how dose HCN4 over expression and decreased Ito impact the VF inducibility, PR,PQ and QT in the Discussion section, please clarify. And it is also unclear whether these change of PR, PQ, QG prolongation found in human athlete.

7. In table one only the ECG pattern shows "*" indicating significant difference. But in the manuscript the author mentioned about the difference in echocardiography findings and autopsy. But there is no "*". Please clarify.

---

## [Author Response]

Reviewer #1 (Recommendations for the authors):– I am concerned about the message that elite exercise is associated with ventricular arrhythmogenesis and enhanced risk for fibrillation.

As clearly stated and discussed in the manuscript on several occasions, regular physical activity has a definite beneficial effect on cardiovascular health, including lower risks of heart disease, diabetes, and stroke. In canine models, for example, both acute and chronic exercise prevented acute-ischemia-induced ventricular fibrillation (Bonilla et al., 2012; Holycross et al., 2007) (Babai L et al., 2022). However, considering the U-shaped relationship between exercise intensity and the risk of adverse cardiovascular events (Merghani et al., 2016), an increasing body of evidence demonstrates that beyond an optimal level of exercise, athletes are more prone to develop ventricular arrhythmias than their age-matched control population. This aspect was added into the revised manuscript.

Although in many cases the difference in the incidence of the arrhythmias is relatively modest and statistically hard to reveal or even undetectable, the overall data still consistently suggest a higher incidence of ventricular arrhythmias in elite athletes. For example, Palatini *et al.* observed that the prevalence of ventricular ectopic activity was higher among runners and cyclists than in the control population, even though the difference was not significant (Palatini et al., 1985). Their results were in agreement with those of Talan *et al.,* who also found ventricular ectopic beats in 70 % of the long-distance runners they studied (Talan et al., 1982).

In the 2000s, studies in Italian sports cardiology showed that frequent and complex ventricular ectopic activity documented by 24-hour ambulatory electrocardiographic monitoring (Holter) is not uncommon among elite athletes. Later, Biffi *et al.* examined the relationship between training-induced left ventricular hypertrophy and risk for ventricular tachyarrhythmias among a large population of young elite athletes without any cardiovascular abnormalities (Biffi et al., 2002). They demonstrated that the prevalence of ventricular ectopic beats and tachyarrhythmias were increased in elite athletes. They suspected it was caused by the intense training-induced shift in the autonomic nervous system.

Additionally, there is an increasing body of evidence that athletes develop non-ischemic left ventricular scars. This myocardial lesion has been recently recognized as an arrhythmic substrate in athletes (Biffi et al., 2008).

These earlier outcomes and our findings may provide evidence that the mild repolarization remodeling and the mild non-ischemic left ventricular scar can be associated with an increase in the number of ventricular arrhythmias, which may contribute to the very low incidence of life-threatening arrhythmias and sudden cardiac death events. We are suggesting that exercise at a competitive level may be an additional potential risk factor that should be taken into account to prevent possible complications, where for example, the repolarization reserve is impaired. These may be otherwise harmless underlying conditions, such as anti-inflammatory drugs or electrolyte imbalance due to excessive sweating.

References:

Babai L, Szigeti Z, Parratt JR, and Végh A. Delayed cardioprotective effects of exercise in dogs are aminoguanidine sensitive: possible involvement of nitric oxide. Clin. Sci. (Lond). 2002;102(4):435–445.

Biffi, A., Maron, B. J., Di Giacinto, B., Porcacchia, P., Verdile, L., Fernando, F., Spataro, A., Culasso, F., Casasco, M., and Pelliccia, A. (2008). Relation between training-induced left ventricular hypertrophy and risk for ventricular tachyarrhythmias in elite athletes. *Am J Cardiol*, *101*(12), 1792-1795. https://doi.org/10.1016/j.amjcard.2008.02.081

Biffi, A., Pelliccia, A., Verdile, L., Fernando, F., Spataro, A., Caselli, S., Santini, M., and Maron, B. J. (2002). Long-term clinical significance of frequent and complex ventricular tachyarrhythmias in trained athletes. *J Am Coll Cardiol*, *40*(3), 446-452. https://doi.org/10.1016/s0735-1097(02)01977-0

Bjornstad, H., Storstein, L., Meen, H. D., and Hals, O. (1991). Electrocardiographic findings in athletic students and sedentary controls. *Cardiology*, *79*(4), 290-305. https://doi.org/10.1159/000174893

Bonilla, I. M., Belevych, A. E., Sridhar, A., Nishijima, Y., Ho, H. T., He, Q., Kukielka, M., Terentyev, D., Terentyeva, R., Liu, B., Long, V. P., Gyorke, S., Carnes, C. A., and Billman, G. E. (2012). Endurance exercise training normalizes repolarization and calcium-handling abnormalities, preventing ventricular fibrillation in a model of sudden cardiac death. *J Appl Physiol (1985)*, *113*(11), 1772-1783. https://doi.org/10.1152/japplphysiol.00175.2012

Holycross, B. J., Kukielka, M., Nishijima, Y., Altschuld, R. A., Carnes, C. A., and Billman, G. E. (2007). Exercise training normalizes β-adrenoceptor expression in dogs susceptible to ventricular fibrillation. *Am J Physiol Heart Circ Physiol*, *293*(5), H2702-2709. https://doi.org/10.1152/ajpheart.00763.2007

Merghani, A., Malhotra, A., and Sharma, S. (2016). The U-shaped relationship between exercise and cardiac morbidity. *Trends Cardiovasc Med*, *26*(3), 232-240. https://doi.org/10.1016/j.tcm.2015.06.005

Palatini, P., Maraglino, G., Sperti, G., Calzavara, A., Libardoni, M., Pessina, A. C., and Dal Palu, C. (1985). Prevalence and possible mechanisms of ventricular arrhythmias in athletes. *Am Heart J*, *110*(3), 560-567. https://doi.org/10.1016/0002-8703(85)90075-4

Talan, D. A., Bauernfeind, R. A., Ashley, W. W., Kanakis, C., Jr., and Rosen, K. M. (1982). Twenty-four hour continuous ECG recordings in long-distance runners. *Chest*, *82*(1), 19-24. https://doi.org/10.1378/chest.82.1.19

– Elite athletes often train for years to reach peak performance. I am wondering what the translational value of 4 months of training in dogs is relative to elite human athletes.

The reviewer is right that elite athletes train longer than 4 months, even years, and this is a limitation of our study. Nevertheless, almost all published data in this field report regular, daily training for 6–12 weeks (Benito et al., 2011). Dog-to-human comparisons are difficult to make accurately, both in terms of exercise tolerance, age, growth, and development, and this is a potential limitation of the model. Yet, the literature suggests that dogs have a similar but accelerated developmental trajectory than humans, including infancy, puberty, adulthood, and ageing, but all of this is only about 20 % of the human lifespan (Wang et al., 2020). This suggests that a training protocol of a few months duration in this dog model would be equivalent to a longer form of training in human studies.

Furthermore, one year of prolonged and intensive endurance training leads to cardiac morphological adaptations in previously sedentary young individuals, similar to those observed in elite endurance athletes (Arbab-Zadeh et al., 2014). Several other studies suggest that even a few months of exercise can have significant cardiovascular effects (Geenen et al., 1982; Schmidt et al., 2014).

In our present study, we used a 16-week high-intensity training protocol that resulted in remarkable cardiac hypertrophy and electrophysiological remodeling. Therefore, it may be assumed that the length of the training protocol applied could provide relevant data on the alteration of the athlete's heart. And, in addition to the undoubted advantages of a more extended dog training protocol, there are also several disadvantages, financial and technical barriers. Following your concern, however, a comment has been included that refers to the potential limitation of our study: “An additional possible limitation is that elite athletes train for several years to reach peak performance, however, in this study only 4 months of intense training could be applied, the exact interspecies translational value of which is difficult to estimate.”

References:

Arbab-Zadeh, A., Perhonen, M., Howden, E., Peshock, R. M., Zhang, R., Adams-Huet, B., Haykowsky, M. J., and Levine, B. D. (2014). Cardiac remodeling in response to 1 year of intensive endurance training. *Circulation*, *130*(24), 2152-2161. https://doi.org/10.1161/CIRCULATIONAHA.114.010775

Benito, B., Gay-Jordi, G., Serrano-Mollar, A., Guasch, E., Shi, Y., Tardif, J. C., Brugada, J., Nattel, S., and Mont, L. (2011). Cardiac arrhythmogenic remodeling in a rat model of long-term intensive exercise training. *Circulation*, *123*(1), 13-22. https://doi.org/10.1161/CIRCULATIONAHA.110.938282

Geenen, D. L., Gilliam, T. B., Crowley, D., Moorehead-Steffens, C., and Rosenthal, A. (1982). Echocardiographic measures in 6 to 7 year old children after an 8 month exercise program. *Am J Cardiol*, *49*(8), 1990-1995. https://doi.org/10.1016/0002-9149(82)90220-x

Schmidt, J. F., Hansen, P. R., Andersen, T. R., Andersen, L. J., Hornstrup, T., Krustrup, P., and Bangsbo, J. (2014). Cardiovascular adaptations to 4 and 12 months of football or strength training in 65- to 75-year-old untrained men. *Scand J Med Sci Sports*, *24 Suppl 1*, 86-97. https://doi.org/10.1111/sms.12217

Wang, T., Ma, J., Hogan, A. N., Fong, S., Licon, K., Tsui, B., Kreisberg, J. F., Adams, P. D., Carvunis, A. R., Bannasch, D. L., Ostrander, E. A., and Ideker, T. (2020). Quantitative Translation of Dog-to-Human Aging by Conserved Remodeling of the DNA Methylome. *Cell Syst*, *11*(2), 176-185 e176. https://doi.org/10.1016/j.cels.2020.06.006

– The enhanced expression of HCN4 in light of reduced heart rate in trained animals is counterintuitive. How do the authors explain this?

The pivotal role of the HCN4 protein in the development of the funny current (I_f_) in the sino-atrial node is well established (D'Souza et al., 2021; Verkerk and Wilders, 2014). However, recent studies have shown that the HCN4 protein, and thus I_f_, is also present in the working myocardium (Lin et al., 2009). Although the role of this transmembrane ionic current in the left ventricle is still controversial, it may contribute to the maintenance of the resting membrane potential of left ventricular myocytes (Stieber et al., 2003). Interestingly, in certain conditions, such as heart failure or hypertrophic cardiac myopathy, a potentially adverse overexpression of HCN4 protein may contribute to the increased arrhythmogenicity observed in these hearts (Cerbai et al., 1997; Cerbai et al., 2001).

In elite athletes, similarly to our observations in this model, left ventricular hypertrophy develops as part of the adaptation to the increased physical demands (Prior and La Gerche, 2012). Even though this type of adaptation is considered physiological, it may share similar features with the hypertrophied heart mentioned above with regard to increased expression of HCN4 protein (Lin et al., 2009).

Based on our immunocytochemical measurements, long-term vigorous endurance training resulted in a significant increase in left ventricular HCN4 protein density in trained animals compared to sedentary subjects. This may have a role in the development of ventricular arrhythmias in the athlete’s heart.

Furthermore, the trained animals exhibited significant resting bradycardia compared to the sedentary dogs. Although it has not been directly investigated, we do not exclude that sino-atrial remodeling following long-term endurance exercise may also play a role in the development of exercise-induced bradycardia. The potentially opposing changes in HCN4 protein expression levels in the pacemaking region and the left ventricle may represent tissue-specific differences in remodeling, however, this requires further investigation.

References:

D'Souza, A., Wang, Y., Anderson, C., Bucchi, A., Baruscotti, M., Olieslagers, S., Mesirca, P., Johnsen, A. B., Mastitskaya, S., Ni, H., Zhang, Y., Black, N., Cox, C., Wegner, S., Bano-Otalora, B., Petit, C., Gill, E., Logantha, S., Dobrzynski, H.,... Boyett, M. R. (2021). A circadian clock in the sinus node mediates day-night rhythms in Hcn4 and heart rate. *Heart Rhythm*, *18*(5), 801-810. https://doi.org/10.1016/j.hrthm.2020.11.026

Cerbai, E., Pino, R., Porciatti, F., Sani, G., Toscano, M., Maccherini, M., Giunti, G., and Mugelli, A. (1997). Characterization of the hyperpolarization-activated current, I(f), in ventricular myocytes from human failing heart. *Circulation*, *95*(3), 568-571. https://doi.org/10.1161/01.cir.95.3.568

Cerbai, E., Sartiani, L., DePaoli, P., Pino, R., Maccherini, M., Bizzarri, F., DiCiolla, F., Davoli, G., Sani, G., and Mugelli, A. (2001). The properties of the pacemaker current I(F)in human ventricular myocytes are modulated by cardiac disease. *J Mol Cell Cardiol*, *33*(3), 441-448. https://doi.org/10.1006/jmcc.2000.1316

Lin, H., Xiao, J., Luo, X., Chen, G., and Wang, Z. (2009). Transcriptional control of pacemaker channel genes HCN2 and HCN4 by Sp1 and implications in re-expression of these genes in hypertrophied myocytes. *Cell Physiol Biochem*, *23*(4-6), 317-326. https://doi.org/10.1159/000218178

Prior, D. L., and La Gerche, A. (2012). The athlete's heart. *Heart*, *98*(12), 947-955. https://doi.org/10.1136/heartjnl-2011-301329

Stieber, J., Herrmann, S., Feil, S., Loster, J., Feil, R., Biel, M., Hofmann, F., and Ludwig, A. (2003). The hyperpolarization-activated channel HCN4 is required for the generation of pacemaker action potentials in the embryonic heart. *Proc Natl Acad Sci U S A*, *100*(25), 15235-15240. https://doi.org/10.1073/pnas.2434235100

Verkerk, A. O., and Wilders, R. (2014). Pacemaker activity of the human sinoatrial node: effects of HCN4 mutations on the hyperpolarization-activated current. *Europace*, *16*(3), 384-395. https://doi.org/10.1093/europace/eut348

– Did the authors collect serum or plasma samples to examine neurohormonal markers? This would aid the autonomic hypothesis postulated in the study which are currently unsupported.

Yes, however, additional studies are currently being carried out in our laboratory, and we plan to test the samples as a whole once the correct number of samples has been reached.

Reviewer #2 (Recommendations for the authors):Thank you for the opportunity to review this manuscript. In general, I feel that this original contribution is one that makes a great start, and the authors are to be congratulated for the study. Much of the recommendations could be revised and further details are as below. There are some methodological issues detailed – I think these should be listed in the limitations. Given that there is very limited data here- I do not feel that these limitations should preclude publication.I do feel that the scientific data comprising the evidence for ventricular arrhythmogenesis in exercise training would benefit a review -and it may be worthwhile (once revisions have taken place to a satisfactory level) to consider an editorial commentary (which I would be happy to assist with).1. I would suggest paying attention to the abstract and its format -I cannot see any information to guide authors on the eLife submission guide; but typically, it would be best formatted into a clear background, methods, results, and conclusion section.

Thank you for the reviewer’s comments. We have improved the formatting of the manuscript abstract.

2. I agree that smaller animal models less helpful – than in canines, however, given this study also has limitations in extrapolation to humans- I would suggest the removal of the statement: Page 2 lines 39-41 (abstract). The limitation of using animal models has been well-discussed in a very good limitations section.

Thank you for this comment, the statement has been removed from the abstract.

3. Present the values in the abstract in order of mention (lengthened depol = higher value then lower) etc throughout.

Thank you for the comment, in the abstract now we are presenting the values in the order of mentioning.

4. I feel that the whole manuscript requires careful editing to ensure that each section is explained clearly, and in order, to avoid any confusing sections. Another reader who is less familiar with the experiments would greatly help. I do not believe that this is a major issue- the paper is generally written reasonably, and the subheadings work well.a. For instance- methods. The section on animals being euthanised can appear last after the in-vivo methods described first. The morphology should be second-last.

Thank you for the reviewer’s recommendations. The methods section has been rearranged according to the experimental protocol so that open-chest experiments are followed by euthanasia of the animals, then the cardiac morphological parameters, and finally the in vitro experiments are described. A new figure and table with an experimental timeline is also presented for better understanding (Figure 1, Table 1).

b. The discussion also needs careful revision to avoid repetition and to make sure that the discussion appropriately places the work in the current scientific context.

Thank you for the reviewer’s constructive comments, we have carefully reviewed and, where necessary, corrected the discussion for better understanding.

c. The conclusion also requires shortening to contain a crux summary statement (or two).

The conclusion has been shortened and re-written.

d. The limitations section is well written.

Thank you for this comment.

e. The conclusion needs to be shorter – much of this should be in the discussion. I do like the authors statement on the clearly beneficial effects of exercise -and these apply only to a subset. This should, however, be in the discussion. The conclusion should really be 2 sentences at most.

The conclusion has been shortened.

f. The discussion should ideally begin with a summary of the principal findings and then move to the discussion points.

Thank you for the reviewer’s suggestion. The discussion starts as required.

5. Introduction: the evidence behind AF and VA is well-understood. It may be better to introduce the "U-shaped" curve of the benefits of exercise with lower levels clearly bad and higher levels -associated with AF.a. Then as you have done, proceed with the population and mechanism data linking exercise training with VA/SCD.

The proposed additions have been added to the manuscript. (Merghani et al., 2016)

Reference:

Merghani, A., Malhotra, A., and Sharma, S. (2016). The U-shaped relationship between exercise and cardiac morbidity. *Trends Cardiovasc Med*, *26*(3), 232-240. https://doi.org/10.1016/j.tcm.2015.06.005

6. Introduction line 82- please summarise the cardiac dysfunction associated with high intensity exercise

In the revised manuscript we have supplemented the Introduction section: “Also, in a recent study in rats after high intensity chronic exercise, it was found that exercise trained rats developed eccentric cardiac hypertrophy, together with both left ventricular (decreased S-wave in pulmonary vein flow and increased left ventricular isovolumic relaxation time) and right ventricular (decreased E-wave velocity, prolonged E-wave deceleration time) diastolic dysfunction and with atrial enlargement. Also, collagen deposition in the right ventricle was significantly higher, which were associated with enhanced vulnerability to arrhythmia at the supraventricular level (Benito et al., 2011).”

Reference:

Benito, B., Gay-Jordi, G., Serrano-Mollar, A., Guasch, E., Shi, Y., Tardif, J. C., Brugada, J., Nattel, S., and Mont, L. (2011). Cardiac arrhythmogenic remodeling in a rat model of long-term intensive exercise training. *Circulation*, *123*(1), 13-22. https://doi.org/10.1161/CIRCULATIONAHA.110.938282

7. Is there a consensus as to the definition of high-intensity exercise?

Exercise intensity is the amount of energy used during exercise. Most physical activities can be performed at varying intensities, from light to heavy. Training experts assess energy cost in metabolic equivalents, or METs, where MET indicates energy expenditure expressed as a multiple of resting metabolic rate and is explained by functional capacity.

McArdle *et al.* presented a classification system for rating the difficulty of sustained physical activity according to intensity: Light work is considered as that eliciting an energy expenditure of 4 METs or less, when the activity results in only minimal perspiration and only a slight increase in breathing. Activities in the range of 5–8 METs are considered moderate intensity when the activity results in definite perspiration and above normal breathing. Heavy work is defined as that requiring 6–8 METS. Excessively heavy work is defined as any effort requiring an increase in metabolism greater than tenfold above resting value (i.e., 10 METS). Based on the literature, the approximate metabolic costs of running at 15 km/h represented as 14.6 METs. For example, this is a running speed that we have often used in our training protocol. (Jette et al., 1990)

As work intensity is the most important factor in the establishment of a conditioning training program, preliminary research was carried out to adapt the level of training to the conditioning level of the animal during the training period. Although MET is a non-invasive and tool-free method for determining exercise intensity in humans, it is difficult to apply to dogs. Observations of heart rate, ECG parameters, and signs and symptoms of fatigue showed that our training protocol was in the "heavy" range, as our dogs were exhausted and their breathing were above normal. Since there is no literature data available in this field, we could only use our estimations, and this is a potential limitation of this work. Unfortunately, we were not able to take ECG or heart rate measurements during the running sessions, however, the changes observed suggest an effective training level for the model.

Reference:

Jette, M., Sidney, K., and Blumchen, G. (1990). Metabolic equivalents (METS) in exercise testing, exercise prescription, and evaluation of functional capacity. *Clin Cardiol*, *13*(8), 555-565. https://doi.org/10.1002/clc.4960130809

8. What was the conditioning protocol? This should be detailed in the methods.

Based on Reviewer #2's question above, the need for and design of the conditioning protocol has been added to the original manuscript as: “Before starting the experiments, the dogs were conditioned to the training protocol, where they were familiarized with the research personnel and were given a few minutes of continuous walk on the treadmill to minimize the distress during training.”

9. Why were echo parameters such as EF, atrial and ventricular volumes not assessed? Both left and right ventricular volumes would be helpful.

The manuscript focused on morphological changes of the left ventricle, demonstrating the effectiveness of the training and the development of the athlete’s heart. A detailed study of functional cardiac parameters and of the right ventricle is planned in the future. However, some of the mentioned parameters have been evaluated and are shown in the manuscript as “revised Table 1”. Ejection fraction, left atrial and left ventricular volumes have been attached. The Methods and Results sections have been supplemented with the corresponding descriptions.

10. Why was fibrosis not quantitatively measured?a. This is a methodological limitation.

At the Department of Pathology, there was no opportunity to analyze myocardial fibrosis quantitatively. As described by Mustroph et al., quantitative analysis of fibrosis can be based on an appropriate software measuring the amount of fibrotic area per total area on digitized slides. Such software was not available during the evaluation. This is a limitation of the study; however, the semi-quantitative assessment in histology reports is widely accepted in human pathology. (Mustroph et al., 2021).

Reference:

Mustroph, J., Hupf, J., Baier, M. J., Evert, K., Brochhausen, C., Broeker, K., Meindl, C., Seither, B., Jungbauer, C., Evert, M., Maier, L. S., and Wagner, S. (2021). Cardiac Fibrosis Is a Risk Factor for Severe COVID-19. *Front Immunol*, *12*, 740260. https://doi.org/10.3389/fimmu.2021.740260

11. Open chest procedures- the protocol needs to be detailed further.a. How was access specifically obtained?b. What was used to produce burst pacing and where in the ventricle was the catheter? Or electrode placed? Was pacing performed at a single site or two sites on the ventricle? Why was the timing 1-3 to 6-9 seconds? Were there between animal differences? Did this produce differences between the groups in terms of pacing protocol?

a., b: We further detailed the surgical description in the revised manuscript and thus the access in the methodology: Pacing was performed via a left thoracotomy from the epicardium using active electrodes placed in the apex of the left ventricle, in all animals and for both protocols. During the arrhythmia provocation, we used 4 different durations of pacing with consecutively increasing durations at 800/min frequency. The stimuli of different durations were necessary because the shorter the duration of the stimulation, the more sensitive and arrhythmogenic the myocardium is, because the longer the vulnerable period, the more likely it is that even a short stimulus falls into the vulnerable period and provokes an arrhythmia. The two groups can thus also be compared from the point of view of how long the stimulus caused arrhythmias. The durations were determined so that even the shortest duration covered at least 1 RR interval.

i. A transvenous catheter would have been better for VF induction.

Indeed, even in clinical practice, arrhythmia induction for diagnostic purposes is performed using transvenous electrodes. However, the hearts of these animals were used for "in vitro" experiments, and thoracotomy had to be used to remove the heart, so the electrode implanted from the epicardium was technically simpler and took much less time than the implantation of the transvenous electrode.

ii. A more refined protocol for VT stimulation with different sites, burst and programmed extrastimuli ((PES) for set durations, at varying CL) and PES at varying coupling intervals down to VERP would have been much better.

Thank you for the reviewer’s comment. Programmed extra stimulation would indeed have been a more appropriate protocol, however, the Effecta D pacemakers in our possession are not capable of performing programmed stimulation.

12. Statistical analysis – was the data checked for normal distribution? Normality of the data should be assessed, and the appropriate comparison chosen for statistical analysis. Please clarify.

Unpaired Student's t-test was applied to estimate whether there was a statistically significant difference between the means of sedentary and trained groups. The t-test is described as a robust test with respect to the assumption of normality. This means that a slight deviation from normality does not have much influence when group sizes do not differ very much from each other. Even so, in agreement with the Reviewer, the normality of our data were assessed using the Kolmogorov-Smirnov test and in case of data sets that do not follow normal distribution, Mann-Whitney U Test was applied instead of Student's t-test. When the data could be described by a discrete variable, chi-square (χ2) test was used, which was added to the Statistics paragraph of the Methods section in the revised manuscript.

13. Discussion line 560- I am finding it extremely difficult to draw corollaries between the work of Volders (assessing hypertrophy and fibrosis) in atrioventricular block vs exercise-related fibrosis. This animal model is a model of Tdp (acquired) but I do not feel that this work is an extension of, or related to the work by Volders et al. The discussion will require some revision.

We thank you for this suggestion. Based on *Reviewer #2* suggestion, to avoid misunderstandings, we have deleted the part of the discussion that made the comparison with the study by Volders *et al.* For a detailed explanation, see question 3 of Reviewer #2.

14. What are the possible remodelling mechanisms for the electrophysiologic changes and VA vulnerability – simply fibrosis? Or related to bradycardia (whatever the mechanism)? Or is there any other possible mechanism?a. Autonomic remodelling and other potentials should be discussed.

We thank you for the reviewer’s suggestion, since this is an important issue on this topic, we have added it to the discussion. The following subtitle and content are added to the discussion: “Possible mechanisms of arrhythmias in the canine athlete’s heart model“.

In brief, we are suggesting that several mechanisms may be responsible for VA vulnerability, which may be harmful from both the trigger and the substrate sides of arrhythmogenesis. (1) A slower heart rate would in itself result in a longer APD and enhanced dispersion of cardiac repolarization. (2) Furthermore, bradycardia resulting in longer diastolic intervals would increase the chance of spontaneous diastolic depolarization reaching the firing threshold. (3) Data suggest that vigorous exercise induces significant changes in the cardiac conduction system that may support arrhythmogenesis, however, the underlying mechanisms are not yet completely understood. (4) Increase in HCN4 protein expression – which is considered as a potential arrhythmia promoter in the left ventricle – was observed in the left ventricular myocytes, similar to changes observed in the hypertrophic heart and heart failure. The higher incidence of VF in trained hearts also promotes this hypothesis. (5) Increased dispersion of repolarization, resulting in prolonged QTc interval and increased QT interval variability on ECG and prolonged APD duration in vitro*,* which may be partly or entirely attributable to downregulation of the Ito current found in trained canine hearts, may contribute to increased arrhythmia vulnerability. (6) A significantly higher level of fibrosis in the left ventricular muscle of trained dogs is an important, but not the only factor that may increase susceptibility to arrhythmias.

15. Given the risk of right ventricular dilatation and dysfunction in athletes (a genotype elusive variant of "ARVC" has been seen in athletes)- RV volume, function and mass would be useful.a. Do animals with VF susceptibility/PVC have RV features? See work by Heidbuchel, La gerche etc.

As the present experiment focused on morphological changes of the left ventricle, demonstrating the effectiveness of the training and the development of the athlete’s hearts, we did not investigate the right ventricle. Thank you for your valuable review, and we indeed intend to investigate the right ventricle in the future.

Reviewer #4 (Recommendations for the authors):1. Did the dog undergo a gradually increased training load or constant strength of training? Since there are several kinds of athletes with different training strengths. The author should be specified what kind of athlete should be the subject of this animal model.

In our experiment, a complex training protocol was applied, with gradually increasing speed and inclination ranges. The training protocol was tested in preliminary experiments and set at the maximum level which could be performed without distress. (Polyak et al., 2018).

The manuscript has been revised as follows in the Methods section: “During the a 16-week-long training period, TRN animals were trained for 5 days a week for 2×90 minutes per day at speeds of 12–18 km·h ^−1^ (gradually increasing protocol) and with 2×50 minute interval running per day at fixed speeds of 4 and 22 km·h^−1^.”

Although the limitation of the model is that it does not correlate perfectly with any human physical activity, the change in left ventricular architecture suggests that it is close to that of “isotonic exercise” activities (e.g., long-distance running, cycling, and swimming), as both ventricular size and left ventricular mass increased, similar to chronic volume overload.

Reference for the preliminary experiment:

Polyak, A., Kui, P., Morvay, N., Lepran, I., Agoston, G., Varga, A., Nagy, N., Baczko, I., Farkas, A., Papp, J. G., Varro, A., and Farkas, A. S. (2018). Long-term endurance training-induced cardiac adaptation in new rabbit and dog animal models of the human athlete's heart. *Rev Cardiovasc Med*, *19*(4), 135-142. https://doi.org/10.31083/j.rcm.2018.04.4161

2. The ECG presented in this manuscript is referred to VENTRICULAR EXTRASYSTOLE beats, but it should be considered as escape beat after the pause period. It would be problematic if the calculation is based on the escape beats rather than the so-called premature ventricular complex. Is there any references supporting this ventricular beat to be called as extrasystole beats? How does the escape beats increase ventricular arrhythmia inducibility? It is also unclear for the reviewer to understand whether the escape beat is measure at week 12 or 16?

For clarification regarding arrhythmias, a significantly high incidence of ventricular beats was observed in the trained animals, the majority of which were ventricular escape beats. Ventricular premature beats were less frequent in both experimental groups, however, but were still more common in the trained animals.

Well-trained athletes often have a slow heart rates, with occasional sinus pauses and, frequently multiple escape beats. Escape arrhythmia is considered to be a compensatory mechanism caused by increased vagal tone and/or disturbance in the SAN or other parts of the cardiac conduction system. It can also be interpreted as a form of ectopic pacemaker activity that is unveiled by lack of other pacemakers to stimulate the ventricles. In athletes, however, it is generally considered to be a benign ECG pattern that disappears during exercise as the vagal tone decreases.

Thank you for the reviewer’s precious comment. Terms have been corrected where necessary in the manuscript to avoid misunderstandings and now clearly indicated in the revised manuscript in Figure 4C. Although, via different mechanisms, both forms of beats are important during exercise induced remodeling, their possible cardiological consequences have been described in detail previously.

All ECG parameters were measured at week 16, including the escape beats. The incorrect data in the publication has been corrected as follows: “In conscious dogs, ECG recordings were measured using precordial leads at 0 and at 16 weeks.”

3. The ECG was measure after 10 mins after recording. What are the dogs doing at that time. Since the activity will influence the HR activity and ECG.

The animals were conditioned to the experimental setting prior to the ECG, so they lay still on their abdomens while the ECG was being recorded, under the supervision of a researcher they were familiar with. The animals received both verbal and nutritional rewards during and after the experiment.

4. Are these all dogs having the same ventricular extrasystole burden? It would be possible to show the readers the difference between dogs with VF induction and dogs without VF induction.

Not all animals experienced escape beats or extra beats, but those that did, had a cumulative pattern. It is thought that the observed rate in human athletes may be similar, as not all athletes develop ECG abnormalities, but the difference is significant between the athlete and the non-athlete population. VF induction was tested in 10 – 10 animals from each group. In these animals, we did not observed relationship between either escape or premature beat frequency and the occurrence of ventricular fibrillation.

5. Since the baseline heart rate is different, is it fair to use the pacing rate to compare the VF inducibility? Or should consider use different pacing rate base on the baseline heart rate? please comment.

During the experiments, we did not modify the basic heart rate by using pacemakers, although it is a fact that the heart rate affects the inducibility of arrhythmias (e.g., bradycardia dependent tachyarrhythmias). However, resting bradycardia is part of cardiovascular adaptation to regular exercise, which may play a role in an increased arrhythmia susceptibility. Since we were interested in whether the athlete's heart is more arrhythmogenic than the untrained healthy heart, we did not consider it desirable to artificially influence the heart rate.

6. It is unclear for the reviewer to understand how dose HCN4 over expression and decreased Ito impact the VF inducibility, PR,PQ and QT in the Discussion section, please clarify. And it is also unclear whether these change of PR, PQ, QG prolongation found in human athlete.

Thank you for your questions. For better understanding, we clarified these points in the adequate Discussion sections.

We observed altered ECG parameters in conscious trained animals in our canine model of athlete’s heart due to long-term vigorous training: prolonged PQ, QT, and widened QRS intervals. These changes are not rare among elite athletes. Similar to our results, several authors reported altered ECG parameters as a result of long-term training. They also observed prolonged PQ and QT intervals and, also widened QRS among human athletes compared to non-athletes group (Bjornstad et al., 1991; Sharma et al., 2017).

Presumably, there is an association between altered intrinsic automaticity of the pacemaker region and atrioventricular node and PQ interval prolongation, the slowed conductance and widened QRS interval, and the altered repolarization and prolonged QT interval due to long-term sustained training.

As we answered previously and described in detail in the manuscript, the overexpression of HCN4 protein in the left ventricle in trained dogs may contribute to the higher incidence of ventricular extra beats in conscious animals at the end of the 16-week-long training protocol. We hypothesized that our outcomes share some similarities with other researcher’s studies who observed increased HCN4-related current magnitude in single left ventricular cells in hypertrophied human and animal hearts due to different conditions (Cerbai et al., 1997; Cerbai et al., 2001). Although these conditions are different, similarities cannot be excluded at cellular level.

The decreased I_to_ magnitude may contribute to the prolonged action potential duration in enzymatically isolated left ventricular myocytes and therefore the prolonged QT interval and enhanced QT dispersion in conscious trained animals.

Presumably, the weakened repolarization reserve due to decreased I_to_ magnitude as potential arrhythmia substrate and the overexpression of HCN4 protein as a potential arrhythmia trigger in left ventricular tissue may contribute to the observed increased incidence of arrhythmogenesis and altered ECG in trained animals.

References:

Bjornstad, H., Storstein, L., Meen, H. D., and Hals, O. (1991). Electrocardiographic findings in athletic students and sedentary controls. *Cardiology*, *79*(4), 290-305. https://doi.org/10.1159/000174893

Cerbai, E., Pino, R., Porciatti, F., Sani, G., Toscano, M., Maccherini, M., Giunti, G., and Mugelli, A. (1997). Characterization of the hyperpolarization-activated current, I(f), in ventricular myocytes from human failing heart. *Circulation*, *95*(3), 568-571. https://doi.org/10.1161/01.cir.95.3.568

Cerbai, E., Sartiani, L., DePaoli, P., Pino, R., Maccherini, M., Bizzarri, F., DiCiolla, F., Davoli, G., Sani, G., and Mugelli, A. (2001). The properties of the pacemaker current I(F)in human ventricular myocytes are modulated by cardiac disease. *J Mol Cell Cardiol*, *33*(3), 441-448. https://doi.org/10.1006/jmcc.2000.1316

Sharma, S., Drezner, J. A., Baggish, A., Papadakis, M., Wilson, M. G., Prutkin, J. M., La Gerche, A., Ackerman, M. J., Borjesson, M., Salerno, J. C., Asif, I. M., Owens, D. S., Chung, E. H., Emery, M. S., Froelicher, V. F., Heidbuchel, H., Adamuz, C., Asplund, C. A., Cohen, G.,... Corrado, D. (2017). International Recommendations for Electrocardiographic Interpretation in Athletes. *J Am Coll Cardiol*, *69*(8), 1057-1075. https://doi.org/10.1016/j.jacc.2017.01.015

7. In table one only the ECG pattern shows "*" indicating significant difference. But in the manuscript the author mentioned about the difference in echocardiography findings and autopsy. But there is no "*". Please clarify.

The corresponding significance flags and p values have been added to Table 1.